# PROGRESSIVE KNOWLEDGE DISTILLATION (PKD): A MODULAR APPROACH FOR ARCHITECTURE-AGNOSTIC KNOWLEDGE DISTILLATION

## ABSTRACT

**Knowledge distillation (KD)** is a key technique for training **lightweight deep neural networks**, particularly in **resource-constrained environments**. While existing KD methods utilize intermediate features to improve student models, they often overlook the proper **alignment between teacher-student layers** and fail to select the most **informative data** for training each student layer. These limitations are especially pronounced in **architecture-agnostic scenarios**, where different network architectures complicate knowledge transfer.

We propose **PKD**, a **Progressive Knowledge Distillation** framework that progressively aligns teacher and student layers through **feature-based modularization**. Each student module is trained using the most **representative features** from its corresponding teacher module, starting with the shallowest layers and progressively moving to deeper ones. This training method enables efficient, architecture-agnostic knowledge transfer across a variety of model architectures. **Experiments on CIFAR-100 and ImageNet-1K** demonstrate that PKD outperforms baseline models, achieving performance improvements of up to **4.54%** and **6.46%**, respectively, thereby validating its effectiveness in diverse neural network settings.

## 1 INTRODUCTION

Deep neural networks (DNNs) have achieved unprecedented success across various domains, including **computer vision**, **natural language processing**, and **speech recognition**. However, their deployment in **resource-constrained environments**, such as mobile devices and embedded systems, remains challenging due to their large computational and memory requirements. **Knowledge Distillation (KD)**, first introduced by (Hinton et al., 2015), offers a solution by enabling the transfer of knowledge from a large, cumbersome **teacher model** to a smaller, more efficient **student model**, without significantly compromising performance.

While most KD methods focus on transferring knowledge from the **final output layer** (Bucila et al., 2006; Hinton et al., 2015), recent research has demonstrated that incorporating **intermediate layer representations** can greatly improve the performance of the student model (Romero et al., 2015; Zagoruyko & Komodakis, 2017). However, two major limitations persist in existing methods: (1) the **misalignment** between teacher and student layers, particularly in **heterogeneous architectures**, and (2) the suboptimal **selection of informative training data** for specific student layers. These limitations are particularly problematic in **architecture-agnostic distillation**, where knowledge must be transferred between models of different architectures (e.g., convolutional networks and transformers) Hao et al. (2024); Wang et al. (2021).

We propose **Progressive Knowledge Distillation (PKD)**, a novel framework that addresses these challenges by progressively aligning and training student and teacher models in a modular and sequential manner.

PKD introduces the following key innovations:

1. **Progressive Modular and Sequential Training:** PKD leverages a progressive, modular, and sequential training strategy. Instead of training the entire student model at once, PKD divides both the teacher and student networks into **feature-based modules**, which reflect

the hierarchical structure of deep networks. Each module is responsible for learning specific types of features, such as texture, shape, and high-level semantics. The training process in PKD is both **modular** and **sequential**. It begins by training the shallowest student module with the corresponding teacher module, ensuring that the most **representative features** are transferred first. After the shallow layers are aligned, training proceeds progressively to the deeper modules. This **layer-wise training** ensures optimal alignment at each stage, mitigating the risks of parameter conflicts and knowledge loss that occur when training all layers simultaneously.

This sequential mechanism is especially valuable in **architecture-agnostic KD**, where student and teacher networks may have different architectures (e.g., CNNs vs. transformers). By focusing on progressively aligning the modules of each network, PKD enables efficient knowledge transfer across heterogeneous architectures, improving the performance of the student model in a structured manner.

2. **Novel Application of PCA for Module-Specific Features:** PKD also introduces a novel application of **principal component analysis (PCA)** for feature selection during distillation. While PCA is commonly used for dimensionality reduction (Wold et al., 1987; Jolliffe, 2002), we employ it to compute **module-specific features**. By extracting the most representative features from the teacher's modules, PKD ensures that the student model learns the most relevant information at each stage of training, further enhancing the distillation process. This targeted feature selection is critical in **architecture-agnostic settings**, reducing redundancy and ensuring that the student network captures only the most informative data from the teacher (Kornblith et al., 2019a).

**PKD Framework:** By integrating these innovations, PKD provides a flexible, **architecture-agnostic** framework for knowledge distillation. It offers substantial improvements over traditional KD methods that focus on final output logits or train all layers simultaneously. PKD's **modular** and **progressive training** process ensures efficient knowledge transfer, even when the student and teacher networks have fundamentally different architectures. Our experiments on standard benchmarks such as **CIFAR-100** (Krizhevsky et al., 2012) and **ImageNet-1K** (Deng et al., 2009) demonstrate that PKD significantly outperforms baseline methods, with improvements of up to **4.54%** on CIFAR-100 and **6.46%** on ImageNet-1K, validating its effectiveness in real-world settings.

## 2 RELATED WORK AND BACKGROUND

In this section, we explore knowledge distillation, a crucial technique for model optimization, and Centered Kernel Alignment (CKA), a key tool used to measure the similarity between network modules within our framework.

**Knowledge Distillation (KD)** has emerged as a powerful technique for compressing large models by training a lightweight student model to mimic the output logits of a pre-trained teacher model (Tang et al., 2022; Zhang et al., 2019; Phuong & Lampert, 2019; Luan et al., 2019; Tung & Mori, 2019; Zhu & Gong, 2018). The concept was introduced by (Bucila et al., 2006) and later refined by (Hinton et al., 2015). Subsequent research has improved logits-based KD with enhancements such as structural information (Guo et al., 2021; Park et al., 2019), model ensembles (Malinin et al., 2020), and contrastive learning (Tian et al., 2020). For instance, (Touvron et al., 2021) proposed a logits distillation method for training Vision Transformer (ViT) students, while (Huang et al., 2018) introduced relaxed KL divergence loss for teacher-student models with significant capacity gaps.

Beyond logits, intermediate feature-based KD has gained traction. Hint-based distillation was first proposed by (Romero et al., 2015), with extensions such as attention map mimicry (Zagoruyko & Komodakis, 2017) and advanced feature distillation methods (Hao et al., 2022; Zhang et al., 2020; Chen et al., 2021). ViTKD (Yang et al., 2022) explored feature-based KD for ViT models, while VKD (Miles et al., 2024) introduced orthogonal projection and task-specific normalization for transformers. KD-DETR (Wang et al., 2024), designed for DETR-based object detection, tackles distillation point inconsistencies through shared and specialized queries, enabling effective distillation for both homogeneous (DETR-to-DETR) and heterogeneous (DETR-to-CNN) setups. Similarly, (Liu et al., 2022) proposed partially cross-attention and group-wise projectors to align features across architectures, and ACMKD (Ni et al., 2024) leveraged multi-student mutual learning strategies for diverse inductive biases, further refining cross-architecture KD.

KD methods have also found applications beyond conventional domains. For instance, PCA-based KD reduced photorealistic style transfer models by 20x while maintaining quality (Chiu & Gurari, 2022), and Margin-MSE loss enhanced KD for neural ranking architectures (Hofstätter et al., 2020). Recent approaches like RdimKD (Guo et al., 2023) employed dimensionality reduction to improve generality and flexibility in KD frameworks.

Despite these advancements, feature-based KD methods often neglect critical challenges such as aligning teacher-student layer representations and selecting representative data for training specific student layers. These limitations are especially pronounced in heterogeneous architectures, where differing model characteristics complicate hint-based distillation. Addressing these challenges remains an open problem, particularly in ensuring robust and efficient training for diverse architectures.

**Centered Kernel Alignment (CKA)** is a feature similarity measurement allowing different representations dimensions (Cortes et al., 2012; Kornblith et al., 2019b). In our work, we adopt CKA to compare features extracted by different architectures (e.g., CNN, ViT, and MLP) and also different network layers.

Consider $\mathbf{X} \in \mathbb{R}^{n \times d_1}$ and $\mathbf{Y} \in \mathbb{R}^{n \times d_2}$ as features extracted by two different models, where $n$ denotes the mini-batch size, and $d_1$ and $d_2$ represent the feature dimensions of $\mathbf{X}$ and $\mathbf{Y}$, respectively. CKA quantifies their similarity using the following formula:

$$\text{CKA}(\mathbf{K}, \mathbf{L}) = \frac{\mathcal{D}_{\text{HSIC}}(\mathbf{K}, \mathbf{L})}{\sqrt{\mathcal{D}_{\text{HSIC}}(\mathbf{K}, \mathbf{K})\mathcal{D}_{\text{HSIC}}(\mathbf{L}, \mathbf{L})}}, \tag{1}$$

where $\mathbf{L} = \mathbf{X}\mathbf{X}^T$ and $\mathbf{K} = \mathbf{Y}\mathbf{Y}^T$ are Gram matrices of the features, and $\mathcal{D}_{\text{HSIC}}$ is the Hilbert-Schmidt independence criterion (Gretton et al., 2007), a non-parametric independence measure. The empirical estimator of $\mathcal{D}_{\text{HSIC}}$ can be formulated as: $\mathcal{D}_{\text{HSIC}}(\mathbf{K}, \mathbf{L}) = \frac{1}{(n-1)^2} \text{tr}(\mathbf{KHLH})$, where $\mathbf{H}$ is the centering matrix $\mathbf{H}_n = \mathbf{I}_n - \frac{1}{n}\mathbf{11}^T$. In our PKD framework, we utilize CKA for network modularization and modular feature alignment.

## 3 METHOD

This section presents our Progressive Knowledge Distillation (PKD) method. The PKD framework focuses on two core aspects: (1) **network modularization** and (2) **progressive modular feature alignment**. The key contribution of our approach lies in progressively aligning student modules with the corresponding teacher modules based on their learned representations and utilizing module-specific features for effective knowledge transfer.

### 3.1 NOTATIONS AND PROBLEM DEFINITION

Knowledge distillation (KD) traditionally involves training a smaller student model to learn from a larger, pre-trained teacher model. The primary forms of knowledge transfer are logits and intermediate feature representations. In our setting, logits represent the output probabilities across different classes, while features are the internal learned representations across different layers. The objective is to align these features across teacher and student models in a progressive manner. Our framework leverages the following loss function:

$$\mathcal{L}\text{KT} = \lambda \mathbb{E}(\boldsymbol{x}, y) \sim (\mathcal{X}, \mathcal{Y})[\mathcal{D}\text{CE}(\boldsymbol{p^s}, y) + (1 - \lambda)\mathcal{D}\text{KL}(\boldsymbol{p^s}, \boldsymbol{p^t})], \tag{2}$$

where $(\mathcal{X}, \mathcal{Y})$ denotes the sample and class label distribution (Hinton et al., 2015; Park et al., 2019; Tian et al., 2020; Hao et al., 2024). $\boldsymbol{p^s}$ and $\boldsymbol{p^t}$ represent the predictions from the student and teacher models, respectively, and $\mathcal{D}\text{CE}$ denote the cross-entropy loss function, and $\mathcal{D}\text{KL}$ signifies the Kullback-Leibler divergence. $\lambda$ is a hyperparameter that balances between one-hot label $y$ and soft label $\boldsymbol{p^t}$.

### 3.2 INSIGHT

Existing KD methods often assume that both teacher and student models share the same feature hierarchy across layers. This assumption breaks down when dealing with heterogeneous architectures (e.g., CNN as the student and ViT as the teacher). Such differences in architecture or scale often hinder the performance of the student model if the same features are transferred without modification.

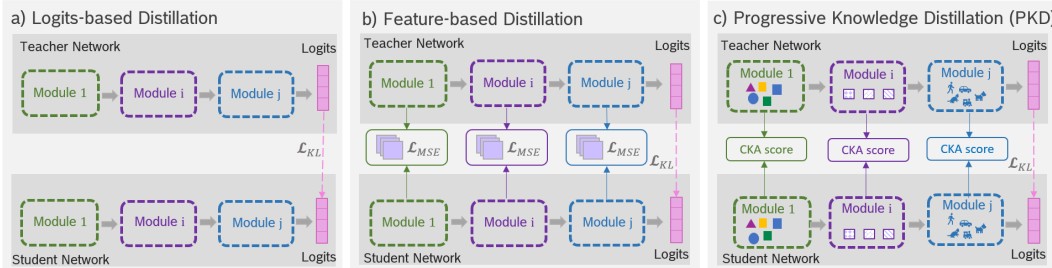

**Figure 1:** Comparison of different KD approaches: (a) Logits-based distillation: the student learns solely from the teacher's final predictions. (b) Feature-based distillation: the student learns from both the final predictions and intermediate features of the teacher. (c) PKD: the student learns *progressively* while aligning its module representations to the corresponding teacher modules, using the most representative data from corresponding teacher module. Only three modules are shown for simplicity; the actual number of modules varies by model.

Our insight is to address this by modularizing the network layers and progressively aligning the student modules with corresponding teacher modules based on **hierarchical representations**.

### 3.3 PKD FRAMEWORK

#### 3.3.1 NETWORK MODULARIZATION

We introduce a **network modularization** technique that organizes network layers into distinct modules based on their **feature similarity**, quantified using **Centered Kernel Alignment (CKA)** scores. Consecutive layers with high CKA scores (i.e., representing similar features) are grouped into the same module, while layers with low CKA scores are assigned to different modules. For any two consecutive layers $i$ and $j$, the **representation distance** $d(i, j)$ is defined as $d(i, j) = 1 - \text{CKA}(i, j)$.

Using the computed CKA scores for all layers, we distinguish two types of representation distances: (1) $d_{SM}$, the distance between consecutive layers within the **same module** (i.e., layers that exhibit similar feature representations), and (2) $d_{DM}$, the distance between layers assigned to **different modules** (i.e., layers that have distinct feature representations). Layers with small distances $d_{SM}$ are grouped into the same module, while layers with significantly larger distances $d_{DM}$ are assigned to different modules. This process is illustrated in **Figure 2**.

In our experiments, we divided the network into three or four modules, depending on its architecture and depth, following this modularization strategy. This process is applied independently to both the student and teacher networks, as their architectures may differ.

**Teacher Network Modularization:** We calculate the CKA score for consecutive layers and group similar layers into the same module. This results in a modular structure where each module captures distinct feature hierarchies.

**Student Network Modularization:** Since the student network is initially untrained, we per-

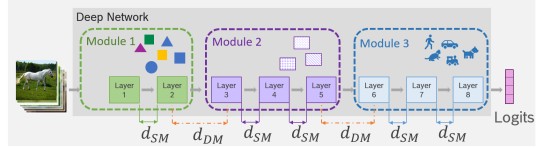

**Figure 2:** Network Modularization Using *CKA* Score: The distance metric (d) is utilized to determine the similarity between network layers. The distance is defined as $d(i, j) = 1 - \text{CKA}(i, j)$, where $i$ and $j$ are the indices of two consecutive layers. A large distance (d) indicates low similarity between layers, guiding the allocation to different modules ($d_{DM}$) for modularization. Conversely, a small distance (d) between consecutive layers indicates shared feature representation, suggesting they belong to the same module ($d_{SM}$).

form a short preliminary training phase (e.g. 5 training epochs) to establish learning patterns. After this phase, we apply the same modularization process using CKA scores to group similar layers into modules.

#### 3.3.2 MODULAR FEATURE ALIGNMENT

This section introduces a novel aspect of our training procedure, focusing on aligning student modules with the most representative features extracted from the corresponding teacher modules. Our key innovation is to align student modules with their corresponding teacher modules based on the teacher's **module-specific features** and utilize these features for effective knowledge distillation, ensuring that the student's learned representations closely match those of the teacher at the module level.

1) **Module-Specific Features** As discussed in Section 1, our approach leverages the hierarchical feature representation of the network for knowledge distillation by modularizing the network. To

extract the module-specific features from the teacher network, we first perform a forward pass to obtain the features from the penultimate layer of each teacher module. Algorithm 1, illustrated in Figure 3, identifies the most important features, which we refer to as **module-specific features**.

For each teacher module $T_i$, we construct a **representation matrix** $R_i^{ns \times m}$, where $ns$ denotes the number of samples and $m$ represents the feature dimension. We then apply principal component analysis (PCA) yielding principal components (PCA-Components) along with their associated variances. Components with higher variance explain more of the overall variance in the dataset and are therefore more important. For more details regarding PCA, refer to Section H of the appendix. The PCA-Components form a matrix with dimensions $n_{\text{components}} \times n_{\text{features}}$, where $n_{\text{components}}$ represents the number of principal components, and $n_{\text{features}}$ denotes the total number of features ($m$). To evaluate the importance of features, we compute the **mean variance** (mv) of the features across the principal components, as outlined in Equation (3). For a detailed explanation of this calculation, please refer to Appendix, section I.4.

$$\text{mv} = \frac{1}{ns} \sum_{i=1}^{ns} \left| \text{PCA}(R_i^{ns \times m}) \right| \tag{3}$$

This provides a ranking of the features, which we sort in descending order to obtain the **sorted indices** (*SI*). We then select the top $k$ features based on a criterion where the mean variance changes smoothly and the gradient of the mean variance approaches zero, indicating the most representative features. Specifically, we select $k$ such that $|\Delta(mv)_k| \leq \epsilon_{th}$, indicating that the difference in mean variance between the $k$ and $k+1$ features is less than a threshold $\epsilon_{th}$. Here, $\epsilon_{th}$ is a predefined threshold ensuring the diminishing variance of additional features. We calculate $k$ as:

$$SI = \text{argsort}(\text{mv}, \text{descending}), \tag{4}$$

$$k = \min \left\{ k' \in SI \mid |\text{mv}[k'] - \text{mv}[k'+1]| \leq \epsilon_{\text{th}} \right\}. \tag{5}$$

We store the top $k$ indices in a vector denoted as *KI* ($KI = SI[: k]$) and use it to retain the corresponding features of the input samples within the teacher module for training the student modules. The unselected features are masked by setting their corresponding values in the representation matrix to zero as follows:

$$R_i[:, MI] = 0, \text{where} \quad MI = SI[k :]. \tag{6}$$

with *MI* denoting the masked indices. This modified representation retains only the most important features relevant to each module, which are used to train the student modules.

---

**Algorithm 1** Module $i$ specific (MS) Features Computation

---

1: **function** Compute-MS(data, module index)
2: i ← module index
3: pca ← PCA()
4: $R_i$ ← Module$_i$(data)
5: pca-components ← pca.fit ($R_i$)
      # Aggregate importance (variance) across all principal components.
6: mean-variance ← Mean(abs(pca-components), axis=0)
      # Sort the features based on their importance.
7: sorted-mean-variance ← Sort(mean-variance, descending)
8: sorted-indices ← argsort(mean-variance, descending)
9: k ← diminish-variance(sorted-mean-variance, threshold)
      # Get k important indices.
10: KI ← sorted-indices[:k]
11: MI ← [KI]'
12: $R_i$[:, MI] ← 0
13: **return** $R_i$
14: **end function**

---

2) **Progressive Modular Alignment** After identifying the module-specific features for each teacher module, the next step is to train the corresponding student modules using these features.

To address this, we employ a progressive approach that systematically selects each teacher module and provides its module-specific data to the student network. This process begins with the shallowest

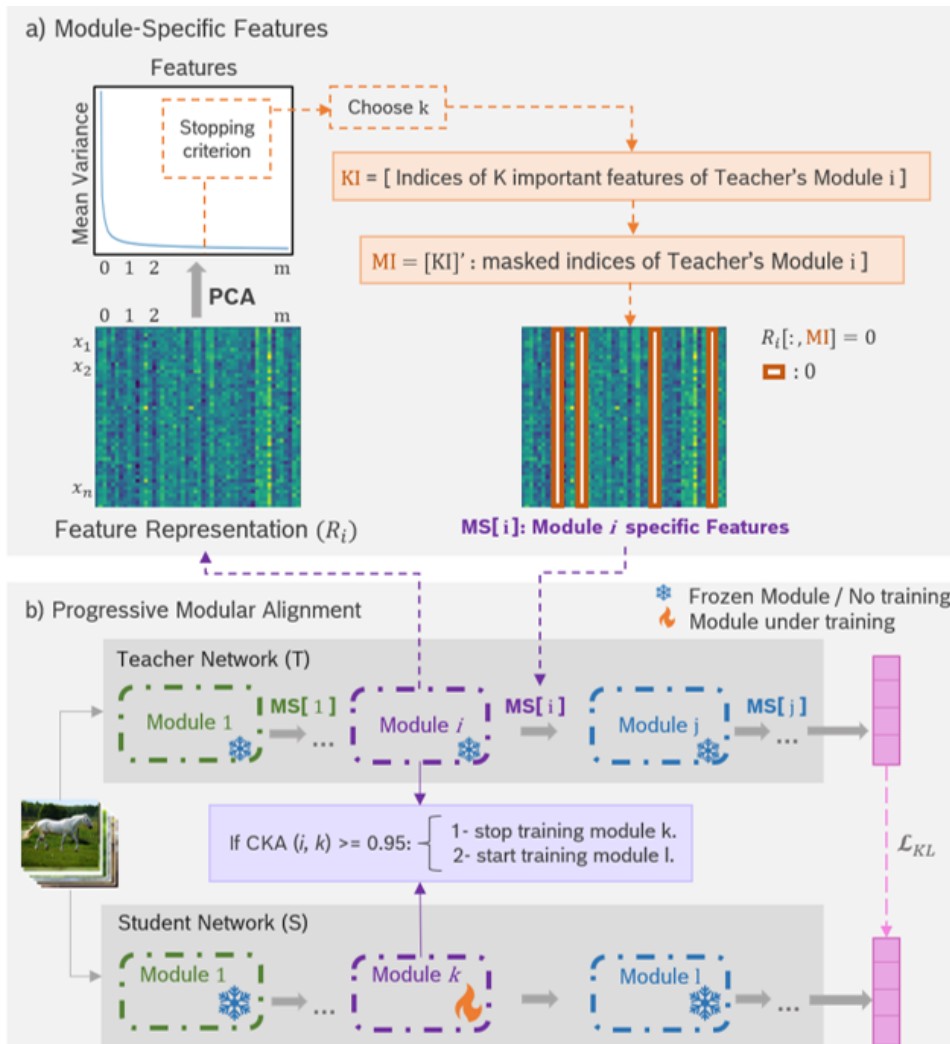

**Figure 3:** This figure illustrates the two main procedures of our method: (a) Module-specific feature computation, where PCA is used to extract the most important features of the teacher module, masking non-important features with zeros. (b) Progressive modular alignment, where each student module $i$ is trained with the corresponding teacher module $i$, using the most representative data computed in (a).

teacher module and gradually moves to deeper modules. The key challenge here is to assign the appropriate student modules for distillation, ensuring that the student modules learn effectively from their corresponding teacher counterparts.

**Module Selection via Binary Vector** We assume the student network (S) consists of $s$ modules. To manage the distillation process, we introduce an $s$-dimensional binary vector $c \in C = \{0, 1\}^s$, where each element $c^{(m)}$ determines whether the $m$-th student module participate in distillation process or remains inactive (frozen). For example, a binary vector $c = [1, 1, 0, 0]$ means that student modules 1 and 2 are active in distillation while modules 3 and 4 remain frozen. For each teacher module $i$, we compute its most specific representation $MS[i]$ as detailed in algorithm 2. By leveraging this representation, we pass the corresponding data through the forward phase of the teacher network. The student module is then trained under various configuration of the binary vector $c$ for a few epochs. This allows us to explore different configurations and select the best corresponding student modules.

**Optimizing Module Representations** Our objective is to discern the configuration $c$ that maximizes the **representation distance** between consecutive *student* modules. This is important to ensure that student modules develop **unique and diverse representations**, minimizing overlap in the knowledge learned. The distance between two successive modules $i$ and $i + 1$ is computed as $d = 1 - \text{CKA}(i, i + 1)$ where CKA is the Centered Kernel Alignment score, which measures the similarity between the feature representations of two modules. By maximizing the representation distance, we ensure that each student module captures distinct aspects of the teacher's knowledge.

**Cross-Validation and Student Configuration Vector (SCV)** To determine the optimal binary vector configuration, we use cross-validation to compute the vector ($c$) that maximizes the average distance between consecutive student modules. This is calculated as:

$$\text{SCV} = \arg\max_{c \in C} \left( \frac{1}{s-1} \sum_{i=1}^{s-1} d_i \right). \tag{7}$$

where $d_i$ is the distance between consecutive student modules $i$ and $i+1$, and $s$ is the total number of student modules. Once the **Student Configuration Vector (SCV)** is identified, it is applied to the student network $S$. Modules that are indicated by 1 in the SCV undergo optimization, while the remaining modules remain frozen:

$$S' = \text{Freeze}(S[m]) \quad \text{if} \quad \text{SCV}[m] == 0, \quad \forall m \in S. \tag{8}$$
$$\text{where} \quad \text{Freeze}(p) : p.\text{requires\_grad} = \text{False}. \tag{9}$$

This ensures that only the selected student modules are optimized, preserving the modular structure of the network.

**Final Training and CKA Convergence** Through this selective optimization process, each student module is aligned with its corresponding teacher module using the **module-specific data**. The training continues until the **CKA similarity** score between the representations of the student and teacher modules exceeds a predefined threshold (e.g., 0.90), indicating high similarity and successful alignment.

To achieve this, we compute the **CKA** score between the entire representation of teacher module $i$ and corresponding student modules every few training runs (e.g., every 5 iterations). If the CKA score exceeds the threshold (e.g., 0.90), training of student modules are stopped, as the modules have sufficiently learned from the corresponding teacher module. We then freeze this *student* module/s and proceed to the next teacher module, identify the corresponding student modules, and train them by aligning their representations with the teacher module.

This iterative process is repeated for each module, progressively aligning the student network to the teacher network. This approach ensures that each student module effectively captures the critical knowledge from the corresponding teacher module while maintaining distinct and unique representations across the entire network. For enhanced clarity, this procedure is illustrated in Figure 4b and detailed in Algorithm 2.

---

**Algorithm 2** Progressive Modular Alignment

---

**Require:** Teacher-modules (T), Student-modules (S), Threshold (th)
1: $c \leftarrow$ binary-vector(len(S))
2: $C \leftarrow$ compute-permutations($c$)
3: **for** $i \leq$ len(T) **do**
4:    $MS[i] \leftarrow$ compute-MS(data, $i$)
     # Please see Alogrithm 1 for more details regarding compute-MS.
5:    SCV $\leftarrow \arg\max_{c \in C} \left( \frac{1}{s-1} \sum_{i=1}^{s-1} d_i \right),$
     #where $d_i = 1 - \text{CKA}(i, i+1), k < \text{len}(S)$
6:    $S' \leftarrow$ Freeze(S[m]) if SCV[m] == 0 $\forall m \in S$
     #Freezes student modules **not** corresponding to teacher module $i$
7:    OI $\leftarrow \{ m \in S : \text{SCV}[m] == 1 \}$
     # OI includes student modules indices corresponding to teacher module $i$
8:    **while** $L_{KL}$ and CKA(T[$i$], $S'[OI]$) $\leq th$ **do**
9:      KD (T($MS[i]$), $S'(data)$))
     #The corresponding student modules to teacher module $i$ are trained using teacher-module $i$ specific features.
10:   **end while**
11:   Freeze(S[OI]) # Once the training of the student modules corresponding to teacher module $i$ is completed, those student modules are frozen.
12: **end for**

---

## 4 EXPERIMENT

### 4.1 EXPERIMENTAL SETUP

We conducted extensive experiments to evaluate the proposed PKD framework. This section provides a concise summary of our experimental configurations; further details available in Appendix sections D and E. All experiments were performed on NVIDIA RTX 4090 GPUs. In our experiments, we followed the OFA experiments and included their baselines for consistency. For a fair comparison, we focused on KD methods specifically designed for heterogeneous architectures. Although we list self-distillation and homogeneous KD methods in our tables for comprehensiveness, we do not compare directly with them, as they differ significantly from our approach. Additionally, ablation studies are reported in the last section and Appendix section B.

**Architectures** We evaluated a variety of models with heterogeneous architectures. Specifically, our experiments included ResNet (He et al., 2016), MobileNet v2 (Sandler et al., 2018), and ConvNeXt (Liu et al., 2021) as CNN models, ViT (Dosovitskiy et al., 2021), DeiT (Touvron et al., 2021), and Swin (Liu et al., 2021) as ViT models, and MLP-Mixer (Tolstikhin et al., 2021) and ResMLP (Touvron et al., 2021)as MLP models. Notably, when Swin-T was used as the student model, it outperformed several teachers when trained from scratch, such as surpassing Mixer-B/16 on the ImageNet-1K dataset. To enable comparisons with these teacher-student pairings, we used two adapted versions of Swin-T: Swin-Nano and Swin-Pico, denoted as Swin-N and Swin-P, following (Hao et al., 2024).

**Datasets** For evaluation, we used the CIFAR-100 dataset (Krizhevsky et al., 2012), consisting of 50K training samples and 10K testing samples with a resolution of 32x32, and the ImageNet-1K dataset (Deng et al., 2009), which contains 1.2 million training samples and 50,000 validation samples with a resolution of 224x224. Since ViTs and MLPs require image patches as input, we upsampled CIFAR-100 images to a resolution of 224x224 to facilitate the patch embedding process.

**Baselines** We employed both logits-based and hint-based KD techniques as our baselines. The hint-based methods included FitNet (Romero et al., 2015), CC (Peng et al., 2019), RKD (Park et al., 2019), and CRD (Tian et al., 2020), while the logits-based baselines included KD (Hinton et al., 2015), DKD (Zhao et al., 2022), DIST (Huang et al., 2022), VKD (Miles et al., 2024), ViTKD (Yang et al., 2022) and OFA (Hao et al., 2024).

**Optimization** In our implementation, all CNN students were trained using the SGD optimizer, while those with ViT or MLP architectures were trained using the AdamW optimizer. For the CIFAR-100 dataset, all baseline models were trained for 300 epochs, whereas our PKD method required only 225 epochs. For the ImageNet-1K dataset, CNN-based baselines were trained for 100 epochs, and ViT and MLP-based baselines were trained for 300 epochs. In contrast, we trained PKD for 75 epochs on CNN-based models and 240 epochs on ViT and MLP-based models. This demonstrates the efficiency of our method, as it requires fewer training epochs, thereby offsetting the computational costs associated with PCA and modularization.

**PCA Computation and Sensitivity Analysis** In PKD, PCA is computed only for the teacher modules, as a pre-processing step before training. Since the teacher network is pretrained, PCA is performed once per module to identify the most informative masked indices, which are then used during student training. Using just 25 randomly selected samples per class was sufficient for accurate index selection, and with four modules, only four PCA computations were required, ensuring computational efficiency. The variance threshold ($\epsilon_{th}$) is set to $1 \times 10^{-4}$. To evaluate the robustness of our approach, we repeated the ImageNet and CIFAR experiments five times, respectively, using different random sample selections in each run. We report the average accuracy and standard deviation, demonstrating the stability of our method despite variations in feature selection.

### 4.2 DISTILLATION RESULTS ON IMAGENET-1K

We first conducted experiments on the ImageNet-1K dataset. To provide a comprehensive comparison, we used five student models and five teacher models from three distinct architecture families, evaluating KD methods across fifteen different teacher-student pairings. The results are presented in Table 1.

Compared to previous KD methods, our PKD framework exhibits significant performance improvements. For CNN-based students, our approach yields gains ranging from $0.61\%$ to $6.46\%$ over the second-best baseline methods. Similarly, for ViT and MLP-based students, our PKD framework achieves substantial accuracy improvements, with maximum gains of $3.21\%$. While OFA (Hao

| Teacher | Student | From Scratch | | hint-based | | | Logits-based | | | | | | | | |
|---|---|---|---|---|---|---|---|---|---|---|---|---|---|---|---|
| | | T. | S. | FitNet | CC | RKD | CRD | KD | DKD | DIST | VKD | ViTKD | NKD | OFA | PKD |
| *CNN-based students* | | | | | | | | | | | | | | | |
| DeiT-T | ResNet18 | 72.17 | 69.75 | 70.44 | 69.77 | 69.47 | 69.25 | 70.22 | 69.39 | 70.64 | 69.33 | 68.51 | 69.81 | 71.34 | 71.97 ± 0.23 |
| Swin-T | ResNet18 | 81.38 | 69.75 | 71.18 | 70.07 | 68.89 | 69.09 | 71.14 | 71.10 | 70.91 | 69.57 | 71.19 | 71.23 | 71.85 | 78.31 ± 0.31 |
| Mixer-B/16 | ResNet18 | 76.62 | 69.75 | 70.78 | 70.05 | 69.46 | 68.40 | 70.89 | 69.89 | 70.66 | 69.14* | 70.08* | 70.21* | 71.38 | 75.81 ± 0.39 |
| DeiT-T | MobileNetV2 | 72.17 | 68.87 | 70.95 | 70.69 | 69.72 | 69.60 | 70.87 | 70.14 | 71.08 | 71.31* | 71.26* | 71.34* | 71.39 | 72.00 ± 0.17 |
| Swin-T | MobileNetV2 | 81.38 | 68.87 | 71.75 | 70.69 | 67.52 | 69.58 | 72.05 | 71.71 | 71.76 | 71.12* | 71.37* | 71.85* | 72.32 | 74.78 ± 0.29 |
| Mixer-B/16 | MobileNetV2 | 76.62 | 68.87 | 71.59 | 70.79 | 69.86 | 68.89 | 71.92 | 70.93 | 71.74 | 71.24* | 70.98* | 71.13* | 72.12 | 73.96 ± 0.36 |
| *ViT-based students* | | | | | | | | | | | | | | | |
| ResNet50 | DeiT-T | 80.38 | 72.17 | 75.84 | 72.56 | 72.06 | 68.53 | 75.10 | 75.60 | 75.13 | 68.51 | 75.22 | 75.31 | 76.55 | 78.08 ± 0.19 |
| ConvNeXt-T | DeiT-T | 82.05 | 72.17 | 70.45 | 73.12 | 71.47 | 69.18 | 74.00 | 73.95 | 74.07 | 69.27 | 74.10 | 74.26 | 74.41 | 77.62 ± 0.10 |
| Mixer-B/16 | DeiT-T | 76.62 | 72.17 | 74.38 | 72.82 | 72.24 | 68.23 | 74.16 | 72.82 | 74.22 | 69.11* | 73.98* | 74.05* | 74.46 | 75.09 ± 0.21 |
| ResNet50 | Swin-N | 80.38 | 75.53 | 78.33 | 76.05 | 75.90 | 73.90 | 77.58 | 78.23 | 77.95 | 77.06* | 77.19* | 77.42* | 78.64 | 79.13 ± 0.29 |
| ConvNeXt-T | Swin-N | 82.05 | 75.53 | 74.81 | 75.79 | 75.48 | 74.15 | 77.15 | 77.00 | 77.25 | 77.39* | 77.21* | 77.13* | 77.50 | 80.43 ± 0.11 |
| Mixer-B/16 | Swin-N | 76.62 | 75.53 | 76.17 | 75.81 | 75.52 | 73.38 | 76.26 | 75.03 | 76.54 | 76.18* | 76.12* | 76.19* | 76.63 | 76.58 ± 0.08 |
| *MLP-based students* | | | | | | | | | | | | | | | |
| ResNet50 | ResMLP-S12 | 80.38 | 76.65 | 78.13 | 76.21 | 75.45 | 73.23 | 77.41 | 78.23 | 77.71 | 74.62 | 78.23 | 77.54 | 78.53 | 79.83 ± 0.32 |
| ConvNeXt-T | ResMLP-S12 | 82.05 | 76.65 | 74.69 | 75.79 | 75.28 | 73.57 | 76.84 | 77.23 | 77.24 | 73.84 | 73.97 | 77.28 | 77.53 | 80.29 ± 0.20 |
| Swin-T | ResMLP-S12 | 81.38 | 76.65 | 76.48 | 76.15 | 75.10 | 73.40 | 76.67 | 76.99 | 77.25 | 74.26* | 74.31* | 75.04* | 77.31 | 79.95 ± 0.38 |

Table 1: Comparison of KD methods across different teacher-student architecture pairings on the ImageNet-1K dataset. Our proposed PKD framework demonstrates significant performance improvements, outperforming all baseline methods. Results marked with * are generated using the authors' provided code. The results for the other baselines are sourced directly from their respective original papers or Hao et al. (2024).

| Teacher | Student | From Scratch | | hint-based | | | Logits-based | | | | | | | | |
|---|---|---|---|---|---|---|---|---|---|---|---|---|---|---|---|
| | | T. | S. | FitNet | CC | RKD | CRD | KD | DKD | DIST | VKD | ViTKD | NKD | OFA | PKD |
| *CNN-based students* | | | | | | | | | | | | | | | |
| Swin-T | ResNet18 | 89.26 | 74.01 | 78.87 | 74.19 | 74.11 | 77.63 | 78.74 | 80.26 | 77.75 | 77.89 | 75.42 | 76.53 | 80.54 | 84.38 ± 0.27 |
| ViT-S | ResNet18 | 92.04 | 74.01 | 77.71 | 74.26 | 73.72 | 76.60 | 77.26 | 78.10 | 76.49 | 76.79 | 77.42 | 78.27 | 80.15 | 84.69 ± 0.31 |
| Mixer-B/16 | ResNet18 | 87.29 | 74.01 | 77.15 | 74.26 | 73.75 | 76.42 | 77.79 | 78.67 | 76.36 | 76.91* | 76.79* | 76.44* | 79.39 | 83.28 ± 0.27 |
| Swin-T | MobileNetV2 | 89.26 | 73.68 | 74.28 | 71.19 | 69.00 | 79.80 | 74.68 | 71.07 | 72.89 | 74.39* | 73.78* | 73.62* | 80.98 | 84.46 ± 0.24 |
| ViT-S | MobileNetV2 | 92.04 | 73.68 | 73.54 | 70.67 | 68.46 | 78.14 | 72.77 | 69.80 | 72.54 | 72.67* | 72.54* | 73.04* | 78.45 | 82.24 ± 0.15 |
| Mixer-B/16 | MobileNetV2 | 87.29 | 73.68 | 73.78 | 70.73 | 68.95 | 78.15 | 73.33 | 70.20 | 73.26 | 73.61* | 73.71* | 74.12* | 78.78 | 81.47 ± 0.22 |
| *ViT-based students* | | | | | | | | | | | | | | | |
| ConvNeXt-T | DeiT-T | 88.41 | 68.00 | 60.78 | 68.01 | 69.79 | 65.94 | 72.99 | 74.60 | 73.55 | 73.09 | 74.17 | 74.11 | 75.76 | 78.32 ± 0.16 |
| Mixer-B/16 | DeiT-T | 87.29 | 68.00 | 71.05 | 68.13 | 69.89 | 65.35 | 71.36 | 73.44 | 71.67 | 71.63* | 71.59* | 73.29* | 73.90 | 75.54 ± 0.31 |
| ConvNeXt-T | Swin-P | 88.41 | 72.63 | 24.06 | 72.63 | 71.73 | 67.09 | 76.44 | 76.80 | 76.41 | 76.44* | 76.37* | 76.86* | 78.32 | 81.26 ± 0.19 |
| Mixer-B/16 | Swin-P | 87.29 | 72.63 | 75.20 | 73.32 | 70.82 | 67.03 | 75.93 | 76.39 | 75.85 | 74.92* | 75.57* | 75.29* | 78.93 | 80.47 ± 0.31 |
| *MLP-based students* | | | | | | | | | | | | | | | |
| ConvNeXt-T | ResMLP-S12 | 88.41 | 66.56 | 45.47 | 67.70 | 65.82 | 63.35 | 72.25 | 73.22 | 71.93 | 72.21 | 72.51 | 73.03 | 81.22 | 84.91 ± 0.27 |
| Swin-T | ResMLP-S12 | 89.26 | 66.56 | 63.12 | 68.37 | 64.66 | 61.72 | - | - | - | 71.89 | 72.82 | 11.05 | 80.63 | 83.88 ± 0.09 |

Table 2: PKD method consistently outperforms the baseline methods, achieving accuracy improvements ranging from 1.50% to 4.54% over the second-best approaches. Comparison of KD methods across different teacher-student architecture pairings on the CIFAR-100 dataset. Results marked with * are generated using the authors' provided code. The results for the other baselines are sourced directly from their respective original papers or Hao et al. (2024).

et al., 2024) performed well as the second-best logits-based method, PKD outperformed all baseline methods across all scenarios, demonstrating its effectiveness in architecture-agnostic distillation.

## 4.3 DISTILLATION RESULTS ON CIFAR-100

In addition to the ImageNet-1K dataset, we evaluated the proposed PKD method on the CIFAR-100 dataset. We conducted experiments with twelve different configurations of teacher and student models, and the results are summarized in Table 2.

On this smaller dataset, hint-based methods demonstrated weaker performance, especially when the student model was based on ViT or MLP architectures, highlighting their limitations in handling heterogeneous architectures. For example, FitNet achieved only 24.06% accuracy when paired with a ConvNeXt-T teacher and a Swin-P student. In contrast, the PKD method consistently outperformed the baseline methods, delivering performance improvements ranging from 1.50% to 4.54% over the second-best approaches.

Interestingly, while DIST ranked third on the ImageNet-1K dataset, DKD often achieved the third-best results on CIFAR-100. This difference stems from the fact that DIST, which relaxes prediction emulation via correlation mimicry, is more effective with a robust teacher model trained on ImageNet-1K. On the other hand, DKD, which amplifies latent knowledge in teacher predictions, is more suited for scenarios involving smaller teachers, such as those trained on CIFAR-100. Despite these variations, our PKD method adaptively enhances target information, enabling the student model to consistently achieve optimal performance. OFA (Hao et al., 2024) secured the second-best performance, with improvements over third-best baseline methods ranging from 0.28% to 8.00%.

## 4.4 ABLATION STUDIES

### 4.4.1 PKD WITHOUT MODULE-SPECIFIC FEATURE COMPUTATION (PKD-PROGRESSIVE)

In this configuration, we progressively trained the student modules without using PCA-based module-specific features, relying on all features instead. As shown in Table 3, while this approach outperforms training from scratch, the performance gains are modest, highlighting the importance of module-specific feature computation for improved knowledge transfer.

### 4.4.2 PKD WITHOUT MODULARIZATION (PKD-PCA)

Here, we skipped modularization and computed PCA features over all data passing through the last layerof the teacher network. These features were used for distillation without splitting the network into modules. As reflected in Table 3, this method shows better performance than PKD-progressive but remains inferior to the fully modularized approach, indicating the critical role of modularization.

### 4.4.3 PKD WITH TWO MODULES (PKD-2 MODULES)

We divided the network into two main modules—the last layer as one module and the rest of the network as the other—and applied PKD. Table 3 shows that this setup performs better than both PKD-progressive and PKD-PCA, but still falls short of the full PKD, underscoring the benefit of finer modularization.

### 4.4.4 FULL PKD

The fully modularized version of PKD achieves the best results across all configurations, as seen in Table 3. This confirms that combining modularization with PCA-based feature computation significantly enhances the knowledge distillation process. For more experiments, please refer to Appendix, section B. In conclusion, the ablation results demonstrate that both modularization and module-specific feature computation are key to achieving optimal performance in PKD.

| Teacher | Student | From Scratch | | Method | | | |
|---|---|---|---|---|---|---|---|
| | | T. | S. | PKD-progressive | PKD-PCA | PKD-2 modules | PKD |
| *CNN-based students* | | | | | | | |
| DeiT-T | ResNet18 | 72.17 | 69.75 | 70.53 | 70.71 | 71.33 | $71.97 \pm 0.23$ |
| Swin-T | ResNet18 | 81.38 | 69.75 | 74.16 | 76.08 | 77.11 | $78.31 \pm 0.31$ |
| Mixer-B/16 | ResNet18 | 76.62 | 69.75 | 71.24 | 71.19 | 73.29 | $75.81 \pm 0.39$ |
| DeiT-T | MobileNetV2 | 72.17 | 68.87 | 71.53 | 71.62 | 71.73 | $72.00 \pm 0.17$ |
| Swin-T | MobileNetV2 | 81.38 | 68.87 | 70.11 | 72.09 | 71.96 | $74.78 \pm 0.29$ |
| Mixer-B/16 | MobileNetV2 | 76.62 | 68.87 | 72.11 | 73.37 | 73.40 | $73.96 \pm 0.36$ |
| *ViT-based students* | | | | | | | |
| ResNet50 | DeiT-T | 80.38 | 72.17 | 76.53 | 76.38 | 77.45 | $78.08 \pm 0.19$ |
| ConvNeXt-T | DeiT-T | 82.05 | 72.17 | 75.36 | 75.12 | 75.93 | $77.62 \pm 0.10$ |
| Mixer-B/16 | DeiT-T | 76.62 | 72.17 | 71.28 | 72.71 | 73.69 | $75.09 \pm 0.21$ |
| ResNet50 | Swin-N | 80.38 | 75.53 | 74.38 | 74.61 | 76.29 | $79.13 \pm 0.29$ |
| ConvNeXt-T | Swin-N | 82.05 | 75.53 | 74.63 | 75.09 | 76.87 | $80.43 \pm 0.11$ |
| Mixer-B/16 | Swin-N | 76.62 | 75.53 | 74.20 | 75.07 | 76.09 | $76.58 \pm 0.08$ |
| *MLP-based students* | | | | | | | |
| ResNet50 | ResMLP-S12 | 80.38 | 76.65 | 78.89 | 78.76 | 79.23 | $79.83 \pm 0.32$ |
| ConvNeXt-T | ResMLP-S12 | 82.05 | 76.65 | 77.74 | 78.39 | 79.08 | $80.29 \pm 0.20$ |
| Swin-T | ResMLP-S12 | 81.38 | 76.65 | 76.81 | 77.37 | 77.98 | $79.95 \pm 0.38$ |

Table 3: Performance comparison of PKD variations on ImageNet-1K. "PKD-progressive" uses all features without module-specific computation, "PKD-PCA" skips modularization, "PKD-2 modules" applies two modules, and "PKD" is the fully modularized approach. Full PKD consistently delivers the best results across all configurations.

## 5 CONCLUSION

This paper presents a novel method for architecture-agnostic knowledge distillation, based on a progressive modular representation framework. Our approach groups network layers with similar feature representations into modules and aligns these representations between the teacher and student networks to facilitate optimal knowledge transfer. By selecting the most representative data from each teacher module for distillation, this method ensures that the student network learns the most relevant information, regardless of differences in architecture. Extensive experiments on CIFAR-100 and ImageNet-1K demonstrate the effectiveness of our method, showing significant improvements in knowledge transfer and student network performance across a wide range of architectures.

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

# A  APPENDIX

This supplementary material provides in-depth information on the following topics:

- More experiments and ablation studies.
- Impact of mini-batch variability on network modularization.
- Architecture of Swin-Nano and Swin-Pico.
- Optimization.
- Related works - Architectures.
- Principal Component Analysis (PCA).
- Principal Component Analysis for Mean Variance Computation.

Each section offers detailed insights into the respective topic for a comprehensive understanding.

# B  MORE EXPERIMENTS AND ABLATION STUDIES

These findings will be included in the final manuscript to highlight the robustness of our method.

## B.1  ABLATION STUDY: COMPARISON WITH ONE-TO-ONE MAPPING

Below is an ablation study comparing our method (**PKD**) with one-to-one mapping (**PKD-1to1**), where teacher modules were identified using CKA scores, and student modules matched their number. All other phases of PKD were kept consistent:

| Teacher | Student | PKD | PKD-1to1 |
|---------|---------|-----|----------|
| **CNN-based students** | | | |
| DeiT-T | ResNet18 | $71.97 \pm 0.23$ | 71.28 |
| Swin-T | ResNet18 | $78.31 \pm 0.31$ | 76.71 |
| DeiT-T | MobileNetV2 | $72.00 \pm 0.17$ | 71.58 |
| **ViT-based students** | | | |
| ResNet50 | DeiT-T | $78.08 \pm 0.19$ | 77.28 |
| ConvNeXt-T | DeiT-T | $77.62 \pm 0.10$ | 76.00 |
| ResNet50 | Swin-N | $79.13 \pm 0.29$ | 76.42 |

Table 4: Comparison of PKD with PKD-1to1, demonstrating the superiority of our modular strategy. Results indicate that PKD consistently outperforms PKD-1to1 across all teacher-student configurations.

The results in Table 4 highlight the benefits of PKD's modular and progressive strategy over the simpler one-to-one mapping approach (PKD-1to1). Specifically:

- CNN-based students: PKD outperforms PKD-1to1 across all teacher-student pairs, with improvements ranging from $+0.42\%$ (DeiT-T $\rightarrow$ MobileNetV2) to $+1.60\%$ (Swin-T $\rightarrow$ ResNet18). This indicates that PKD's progressive alignment enables more effective knowledge transfer, especially in heterogeneous setups.

- ViT-based students: PKD demonstrates even larger gains, such as $+2.71\%$ (ResNet50 $\rightarrow$ Swin-N) and $+1.62\%$ (ConvNeXt-T $\rightarrow$ DeiT-T). This shows that PKD's flexible modularization adapts better to diverse architectures.

The performance gap can be attributed to PKD's ability to align student modules progressively and effectively extract hierarchical representations, compared to the rigid one-to-one mapping in PKD-1to1. PKD also benefits from selectively training student modules while freezing others, which improves efficiency and avoids overfitting.

## B.2 ABLATION STUDY: EVALUATING MODULARIZATION USING FEATURE MAP RESOLUTION

To evaluate the impact of modularization methods, we compared our proposed method (PKD) with an alternative approach that uses **feature map resolution** for modularization, referred to as **PKD-res**. In PKD-res, modules are identified based on the resolution of feature maps at different layers rather than using CKA scores. This method groups layers with the same resolution into the same module.

The experiments were conducted on a variety of teacher-student configurations for CNN, ViT, and MLP architectures across CIFAR-100 and ImageNet-1K datasets. The performance was compared using metrics such as final accuracy and modularization strategy. The results are presented in the following tables.

| Teacher | Student | OFA | PKD | PKD-res |
|---|---|---|---|---|
| **CNN-based students** | | | | |
| DeiT-T | ResNet18 | 71.34 | 71.97 | 70.26 |
| Swin-T | ResNet18 | 72.31 | 78.31 | 71.98 |
| DeiT-T | MobileNetV2 | 71.39 | 72.00 | 71.35 |
| Swin-T | MobileNetV2 | 72.32 | 74.78 | 71.96 |
| **ViT-based students** | | | | |
| ResNet50 | DeiT-T | 76.55 | 78.08 ± 0.19 | 76.08 |
| ConvNeXt-T | DeiT-T | 74.41 | 77.62 ± 0.10 | 75.47 |
| ResNet50 | Swin-N | 78.64 | 79.13 ± 0.29 | 78.91 |
| ConvNeXt-T | Swin-N | 77.50 | 80.43 ± 0.11 | 78.29 |
| **MLP-based students** | | | | |
| ResNet50 | ResMLP-S12 | 78.53 | 79.83 ± 0.32 | 78.94 |
| ConvNeXt-T | ResMLP-S12 | 77.53 | 80.29 ± 0.20 | 78.53 |

Table 5: Teacher modularization based on feature map resolution (PKD-res) compared to OFA and PKD on ViT and MLP-based students.

The results in Tables 5 demonstrates the following key findings:

- **Performance Superiority of PKD:** Across all configurations, PKD consistently outperforms both OFA and PKD-res. This highlights the effectiveness of CKA-based modularization in identifying meaningful layer groupings that facilitate efficient knowledge transfer.

- **Limitations of Feature Map Resolution:** While PKD-res provides a straightforward approach to modularization, it performs worse than PKD in all cases. For instance, in the ResNet50 → DeiT-T configuration, PKD achieves an accuracy of 78.08%, while PKD-res falls to 76.08%. Similarly, for Swin-T → ResNet18, PKD achieves 78.31%, whereas PKD-res reaches only 71.98%.

- **Impact on Modularization Strategy:** The feature map resolution method (PKD-res) groups layers solely based on spatial resolution, which may fail to capture deeper representational similarities critical for effective distillation. In contrast, PKD leverages CKA scores to align features more precisely, leading to better modularization and improved performance.

- **MLP and ViT Architectures:** The performance gap between PKD and PKD-res is more pronounced for MLP and ViT-based students, likely due to the unique feature hierarchies in these architectures that are better captured by CKA-based modularization.

These results underscore the importance of using meaningful similarity metrics like CKA for modularization, as opposed to simpler metrics such as feature map resolution, to achieve superior performance in knowledge distillation.

## B.3 ABLATION STUDY: MISALIGNED MODULARIZATION

To evaluate the impact of modular alignment on knowledge transfer, we conducted experiments with two variations of misaligned modularization compared to our proposed PKD approach:

1. Misaligned Modules: We combined two middle student modules into a single module, creating a mismatch with the teacher's modular structure. This resulted in misaligned modularization, as the

number and composition of student modules did not align with the teacher. 2. Shifted Modules: One module from each group (except the first) was shifted to the previous group. This introduced minor structural shifts in the modularization, testing the robustness of PKD to small changes.

Both approaches were compared with OFA and PKD. The results are presented in Table 6.

| Teacher | Student | OFA | PKD | PKD-Misaligned |
|---|---|---|---|---|
| CNN-based students | | | | |
| DeiT-T | ResNet18 | 71.34 | 71.97 ± 0.37 | 71.54 |
| Swin-T | ResNet18 | 71.85 | 78.31 ± 0.31 | 75.26 |
| ViT-based students | | | | |
| ResNet50 | DeiT-T | 76.55 | 78.08 ± 0.19 | 77.04 |
| ConvNeXt-T | DeiT-T | 74.41 | 77.62 ± 0.10 | 75.83 |
| ResNet50 | Swin-N | 78.64 | 79.13 ± 0.29 | 79.08 |
| ConvNeXt-T | Swin-N | 77.50 | 80.43 ± 0.11 | 78.37 |
| MLP-based students | | | | |
| ResNet50 | ResMLP-S12 | 78.53 | 79.83 ± 0.32 | 78.96 |
| ConvNeXt-T | ResMLP-S12 | 77.53 | 80.29 ± 0.20 | 78.79 |

Table 6: Comparison of teacher-student modularization methods: Aligned (PKD) vs. Misaligned (PKD-Misaligned).

The results in Table 6 reveal the following insights:

1. Aligned Modularization Advantage: PKD consistently outperforms both OFA and PKD-Misaligned across all teacher-student configurations, demonstrating the importance of aligned modularization in facilitating effective knowledge transfer. 2. Impact of Misaligned Modules: Combining two middle student modules led to noticeable performance degradation compared to PKD. For instance, in the Swin-T → ResNet18 configuration, PKD achieves 78.31% accuracy, while PKD-Misaligned drops to 75.26%. 3. Robustness to Minor Shifts: Shifting modules caused no significant accuracy change, suggesting PKD's robustness to minor modularization variations. 4. Performance Retention: Despite misalignment, PKD-Misaligned still outperforms OFA in most configurations, indicating the inherent strength of the PKD framework even when modularization is imperfect.

These findings emphasize the importance of accurate modular alignment in PKD for achieving optimal performance while highlighting its robustness to small modularization shifts.

## B.4 KNOWLEDGE TRANSFER IN CONSISTENT ARCHITECTURES.

We employ the widely recognized pairing of a ResNet34 teacher and a ResNet18 student on the ImageNet-1K dataset to further showcase the efficacy of our method within consistent architectures. As depicted in Table 7, the PKD technique yields results comparable to the top-performing distillation baseline.

| | T. | S. | KD | OFD | Review | CRD | DKD | DIST | OFA | PKD (ours) |
|---|---|---|---|---|---|---|---|---|---|---|
| Accuracy | 73.62 | 69.90 | 70.66 | 70.81 | 71.61 | 71.17 | 71.70 | 72.07 | 72.10 | 73.51 ± 0.18 |

Table 7: KD methods with homogeneous architectures on ImageNet-1K. *T*: ResNet34, *S*: ResNet18.

| Teacher | T. | S.(ResNet50) | RKD | Review | CRD | DKD | DIST | OFA | PKD (ours) |
|---|---|---|---|---|---|---|---|---|---|
| ResNet152 | 82.83 | 79.86 | 79.53 | 80.06 | 79.33 | 80.49 | 80.55 | 80.64 | 82.75 ± 0.30 |
| ViT-B | 86.53 | 79.86 | 79.38 | 79.32 | 79.48 | 80.76 | 80.90 | 81.33 | 84.67 ± 0.22 |

Table 8: Comparison of homogeneous and heterogeneous teacher on ImageNet-1K.

## B.5 CONSISTENT *vs.* DIVERSE TEACHER MODELS

To evaluate the influence of employing a larger diverse teacher model, we train a ResNet50 student with both a ResNet152 teacher (consistent architecture) and a ViT-B teacher (diverse architecture). As illustrated in Table 8, our PKD approach achieves a significant improvement in performance when utilizing the ViT-B teacher compared to the ResNet152 teacher. This finding underscores the importance of architecture-agnostic knowledge distillation (KD) in striving for enhanced performance gains.

### B.6 KD WITH LAST LAYER MODULE-SPECIFIC FEATURES

In this experiment, we utilize the OFA method to examine the impact of using only module-specific features from the last layer, specifically extracting the most important features from this layer. For modularization, we divided the student and teacher networks into four modules, as outlined in the OFA method by (Hao et al., 2024). Initially, we trained the first two modules by leveraging the most important features while keeping the last two modules (the deeper layers) frozen. The rationale is that the shallow layers of the network learn the most critical features and semantics (class-specific features).

After training the first two modules, we proceeded to train the deeper modules using data where the indices corresponding to the most important features were masked by zero. The experimental results, presented in Table 9 labeled as OFA+MS, demonstrate an improvement over the OFA method, highlighting the significance of utilizing the appropriate features to train different layers or modules of the network.

### B.7 ENHANCED DKD PERFORMANCE WITH PKD

We conduct further experiments to assess the effectiveness of our PKD method, particularly the progressive modular alignment, on other knowledge distillation techniques. Specifically, we integrate our method with DKD. In this setup, we adhere to the primary DKD methodology and incorporate modularization along with progressive modular training. The experimental outcomes are presented as (DKD+PKD) in Table 9, highlighted as gray. As shown in the table, our approach enhances the performance of DKD.

| Teacher | Student | From Scratch | | Logits-based | | | | | | |
|---|---|---|---|---|---|---|---|---|---|---|
| | | T. | S. | KD | DKD | DKD +PKD | DIST | OFA | OFA+MS | PKD |
| *CNN-based students* | | | | | | | | | | |
| DeiT-T | ResNet18 | 72.17 | 69.75 | 70.22 | 69.39 | 73.56 | 70.64 | 71.34 | 72.11 | $71.97 \pm 0.23$ |
| Swin-T | ResNet18 | 81.38 | 69.75 | 71.14 | 71.10 | 78.46 | 70.91 | 71.85 | 74.09 | $78.31 \pm 0.31$ |
| DeiT-T | MobileNetV2 | 72.17 | 68.87 | 70.87 | 70.14 | 71.28 | 71.08 | 71.39 | 71.63 | $72.00 \pm 0.17$ |
| *ViT-based students* | | | | | | | | | | |
| ResNet50 | DeiT-T | 80.38 | 72.17 | 75.10 | 75.60 | 78.02 | 75.13 | 76.55 | 77.23 | $78.08 \pm 0.19$ |
| ConvNeXt-T | DeiT-T | 82.05 | 72.17 | 74.00 | 73.95 | 75.94 | 74.07 | 74.41 | 74.83 | $77.62 \pm 0.10$ |
| *MLP-based students* | | | | | | | | | | |
| ResNet50 | ResMLP-S12 | 80.38 | 76.65 | 77.41 | 78.23 | 79.61 | 77.71 | 78.53 | 79.17 | $79.83 \pm 0.32$ |
| ConvNeXt-T | ResMLP-S12 | 82.05 | 76.65 | 76.84 | 77.23 | 80.38 | 77.24 | 77.53 | 76.68 | $80.29 \pm 0.20$ |

Table 9: Applying our PKD method on top of the DKD method improves performance on ImageNet-1K. Applying MS procedures on OFA last layer , OFA+MS, improves its performance.

### B.8 IMPACT OF THE THRESHOLD IN EQUATION 5

To better understand the impact of the threshold parameter $\epsilon_{th}$ in Equation 5 on performance, we conducted an ablation study. This experiment systematically evaluated different values of $\epsilon_{th}$ and their effect on the accuracy of our method. The results of this ablation are presented in Table 10, where we report the performance of the PKD model with varying threshold values.

As shown in Table 10, our method consistently outperforms the OFA baseline across all tested thresholds. Notably, $\epsilon_{th}$ values of $1e-4$ and $5e-4$ yield the best performance, suggesting that fine-tuning the threshold in this range is critical for maximizing model accuracy. These results

| Teacher | Student | PKD considering different $\epsilon_{th}$ | | | | | | | OFA |
|---|---|---|---|---|---|---|---|---|---|
| | | $2e-5$ | $5e-5$ | $1e-4$ | $5e-4$ | $2e-3$ | $5e-3$ | $1e-3$ | |
| DeiT-T | ResNet18 | 71.91 | 72.04 | 71.97 | 71.92 | 71.86 | 71.80 | 71.81 | 71.34 |
| Swin-T | ResNet18 | 78.00 | 78.03 | 78.31 | 78.26 | 78.17 | 78.14 | 78.09 | 71.85 |
| ResNet50 | DeiT-T | 77.86 | 77.99 | 78.08 | 78.03 | 77.91 | 77.83 | 77.80 | 76.55 |

Table 10: Performance comparison of PKD using different threshold values $\epsilon_{th}$ in Equation 5, with varying teacher-student model pairs on the ImageNet dataset. Our method consistently outperforms the OFA baseline across all tested values.

validate the robustness of our approach and demonstrate the importance of selecting an appropriate threshold value for optimal knowledge distillation performance.

## C    IMPACT OF MINI-BATCH VARIABILITY ON NETWORK MODULARIZATION

In our PKD framework, network modularization is not based on individual mini-batches, but rather on a subset of the data to manage computational costs efficiently. Specifically, we randomly select a limited number of samples per class (e.g., 25) for CKA (Centered Kernel Alignment) computations. Through extensive experimentation, we found that this approach produces modularization results comparable to using the entire dataset.

To account for variability when considering all training data, we compute the average CKA score across mini-batches. This ensures consistency in the modularization process while maintaining computational efficiency, as the use of a subset of data for CKA calculation reduces the overall computational burden without sacrificing accuracy in the modularization results.

## D    ARCHITECTURE OF SWIN-NANO AND SWIN-PICO

To ensure that the teacher models surpass the performance of the student model, (Hao et al., 2024) presented two modified versions of Swin-Tiny (Liu et al., 2021), named Swin-Nano and Swin-Pico. Swin-Nano features an embedding dimension of 64, while Swin-Pico has an embedding dimension of 48, in contrast to the original Swin-Tiny's embedding dimension of 96. Additionally, Swin-Tiny has layer depths of (2, 2, 6, 2) and numbers of heads of (3, 6, 12, 24), whereas the two modified models share the same configurations for depths and numbers of heads, which are (2, 2, 2, 2) and (2, 4, 8, 16), respectively.

## E    OPTIMIZATION

For training models with diverse architectures on the ImageNet-1K and CIFAR-100 datasets, we employ distinct optimization settings. The comprehensive settings are provided in Table 11.

|              | ImageNet-1k | | CIFAR-100 | |
|--------------|-------------|-----------|-----------|-----------|
|              | CNN | ViT/ MLP | CNN | ViT/MLP |
| Epochs       | 100 | 300 | 300 | 300 |
| Batch size   | 512 | 1024 | 300 | 300 |
| Initial LR   | 0.1 | 5e-4 | 5e-2 | 5e-4 |
| Minimum LR   | 1e-6 | 1e-6 | 1e-3 | 1e-5 |
| Optimizer    | SGD | AdamW | SGD | AdamW |
| Weight decay | 1e-4 | 5e-2 | 2e-3 | 5e-2 |
| LR schedule  | ×0.1 at [30,60,90] | Cosine | Cosine | Cosine |
| Warmup       | 3 | 20 | 3 | 20 |
| EMA          | - | 0.99996 | - | - |
| RandAugment  | - | 9/0.5 | - | 9/0.5 |
| Mixup        | - | 0.8 | - | 0.8 |
| Cutmix       | - | 1.0 | - | 1.0 |
| RE prob      | - | 0.25 | - | 0.25 |

Table 11: Optimization settings details.

## F    OVERALL PERFORMANCE AND TRAINING EFFICIENCY

### F.1    TRAINING EFFICIENCY

To compare performance and training efficiency, we conducted experiments with **PKD**, **OFA**, and **DKD** under identical setups:

**Key Observations:**

- **Final Performance:** PKD consistently achieves superior accuracy, with improvements such as +6.46% on CIFAR-100 and +1.53% on ImageNet-1K compared to OFA.

| Dataset | Teacher-Student Pair | Method | Final Accuracy (%) | Total Training Time (hrs) | Avg. Time/Epoch (min) | Epochs |
|---|---|---|---|---|---|---|
| CIFAR-100 | Swin-T → ResNet18 | OFA | 71.85 | 14.2 | 2.84 | 300 |
| | | DKD | 71.10 | 14.4 | 2.88 | 300 |
| | | **PKD** | **78.31 ± 0.31** | **11.4** | **3.04** | **225** |
| ImageNet-1K | ResNet50 → DeiT-T | OFA | 76.55 | 110.0 | 22.0 | 300 |
| | | DKD | 75.60 | 112.5 | 22.5 | 300 |
| | | **PKD** | **78.08 ± 0.19** | **74.0** | **18.5** | **240** |

Table 12: Comparison of training efficiency and final performance for PKD, OFA, and DKD under identical setups.

- **Training Budget:** PKD reduces total training time by 20%–30%, thanks to faster convergence and fewer required epochs.
- **Training Efficiency:** PKD's modularization slightly increases time per epoch but achieves overall faster training due to reduced epochs.

## F.2 SCALABILITY CHALLENGES

Although PKD introduces additional steps such as PCA and CKA-based modularization, several design choices address scalability:

- **Subset-Based PCA:** PCA is performed on a small subset (e.g., 25 samples per class), significantly reducing computational overhead while preserving performance.
- **Batch-Based CKA:** Modularization uses averaged mini-batch CKA scores, reducing memory requirements and enabling scalability for larger models.
- **Progressive Training:** Modules are trained sequentially, avoiding simultaneous optimization of the entire network and reducing effective model size during training.

**Proposed Solutions for Enhanced Scalability:**

- **Low-Rank Approximations for PCA:** Techniques like randomized SVD can further reduce computational costs.
- **Approximation Methods for CKA:** Nyström approximations or sketching techniques can compute Gram Matrices efficiently for deeper architectures.
- **Parallel and Distributed Training:** PKD's modular structure is inherently parallelizable, allowing multi-GPU setups.
- **Hierarchical Modularization:** For very deep networks, hierarchical grouping of layers into coarse modules can further reduce complexity.

## F.3 PERFORMANCE GAP AND TRAINING BUDGET

The impact of simplifications in PKD on performance and training budget is summarized below:

- **Subset-Based PCA:** PCA is performed on a small subset of data, leading to:
  - **Performance Gap:** Minimal difference ($\leq 0.2\%$) compared to full-dataset PCA, as shown in ablation studies.
  - **Training Budget Impact:** Reduces preprocessing time by 80%–90%, making it computationally efficient.
- **Variance Threshold ($\epsilon_{th}$):** Features are selected based on their variance, ensuring only the most representative features are retained:
  - **Performance Gap:** Negligible differences ($\leq 0.2\%$) with optimal thresholds ($1e-4$–$5e-4$); extremely low thresholds ($<2e-5$) lead to a slight drop ( 0.4%).
  - **Training Budget Impact:** Reduces feature dimensions, accelerating student training.
- **Batch-Based CKA:** Gram Matrices and CKA scores are computed on mini-batches:
  - **Performance Gap:** Batch-based CKA achieves comparable modularization to full-dataset CKA ($\leq 0.3\%$ difference in accuracy).
  - **Training Budget Impact:** Reduces memory usage and modularization time, enhancing scalability.

**Overall Training Budget Reduction:** PKD reduces total training time by up to 30% due to faster convergence and fewer required computations compared to baseline methods like OFA and DKD.

## G   RELATED WORKS- ARCHITECTURES

In recent years, significant advancements have been made in the evolution of model architectures for computer vision tasks. This section offers a succinct overview of two notable architectures: Transformer and MLP.

**Vision Transformer** Vaswani and colleagues (Vaswani et al., 2017) initially introduced the transformer architecture for tasks in natural language processing (NLP). Due to the utilization of the attention mechanism, this framework adeptly captures prolonged dependencies and attains remarkable performance. Motivated by its considerable success, endeavors have been made to devise transformer-based models for computer vision (CV) tasks. (Dosovitskiy et al., 2021) partition an image into non-overlapping patches and map these patches into embedding tokens. Subsequently, these tokens undergo processing by the transformer model akin to NLP tasks. Their design achieves state-of-the-art performance and stimulates the creation of a sequence of subsequent architectures.

**MLP** For an extended period, MLP has exhibited inferior performance compared to CNN in the domain of computer vision. To explore the potential of MLP, (Tolstikhin et al., 2021) proposed MLP-Mixer, exclusively based on the MLP structure. MLP-Mixer takes embedding tokens of an image patch as input and interleaves channel and spatial information mixing at each layer. This architecture performs comparably to the leading CNN and ViT models. Touvron and colleagues (Touvron et al., 2021) proposed another MLP architecture termed ResMLP.

The most advanced CNN, Transformer, and MLP models achieve analogous performance. Nonetheless, these architectures possess distinct inductive biases, leading to disparate preferences in representation learning. Generally, *noticeable distinctions exist between features acquired by diverse architectures*.

## H   PRINCIPAL COMPONENT ANALYSIS (PCA)

Principal Component Analysis (PCA) is a statistical method aimed at reducing the dimensionality of high-dimensional data while preserving as much variance as possible. This is accomplished by identifying principal components, which are orthogonal vectors that indicate the directions of maximum variance (Shlens, 2014; Jolliffe, 2002).

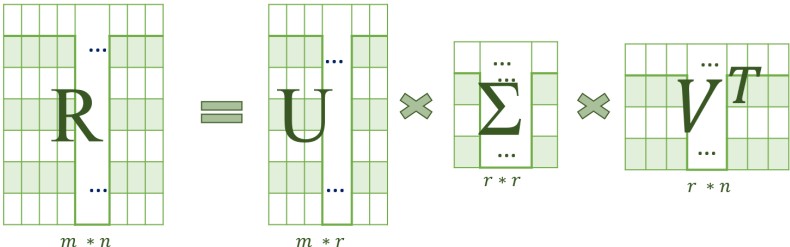

Figure 4: Singular Value Decomposition

### H.1   PROCEDURE

This section details the steps involved in PCA to extract the most informative features.

**Data Standardization:** Standardize the dataset features to have a mean of zero and a unit variance.

**Covariance Matrix:** Calculate the covariance matrix of the standardized data, which captures the relationships between different features.

**SVD of Covariance Matrix:** Perform Singular Value Decomposition (SVD) on the covariance matrix. The SVD of the covariance matrix yields the principal components.

**Selecting Principal Components:** Sort the singular values in descending order. The corresponding singular vectors are the principal components. Select the top $k$ principal components to form a reduced-dimensional space.

**Projection:** Project the original data onto the selected principal components to obtain the lower-dimensional representation.

**Benefits:** - Dimensionality reduction simplifies data visualization and interpretation. - Reduced dimensionality often enhances computational efficiency. - Principal components encapsulate the most significant patterns in the data.

### H.2 MORE EXPLANATION REGARDING SVD COMPUTATION

Consider an $m \times n$ matrix $R$, where $m$ denotes the number of rows and $n$ represents the number of columns. The primary objective of Singular Value Decomposition (SVD) is to decompose matrix $R$ into three distinct matrices: $U$, $\Sigma$, and $V^T$ (the transpose of matrix $V$). This decomposition is expressed as $R = U\Sigma V^T \in \mathbb{R}^{m \times n}$, as illustrated in Figure 4.

- $U$: An $m \times m$ orthogonal matrix, whose columns are the left singular vectors of $R$. - $\Sigma$: An $m \times n$ diagonal matrix containing the singular values of $R$ (non-negative and arranged in descending order).

- $V^T$: An $n \times n$ orthogonal matrix, with columns representing the right singular vectors of $R$.

### H.3 EIGENVALUES AND EIGENVECTORS

Eigenvalues and eigenvectors are also fundamental to understanding matrix properties. An eigenvalue $\lambda$ and its corresponding eigenvector $\mathbf{v}$ of a square matrix $R$ satisfy the equation $R\mathbf{v} = \lambda\mathbf{v}$. Eigenvectors denote directions in the vector space that are scaled by the matrix $R$, while eigenvalues represent the scaling factors for these eigenvectors.

### H.4 SVD AND ITS RELATIONSHIP TO EIGENVALUES AND EIGENVECTORS

SVD establishes a crucial relationship between eigenvalues and eigenvectors and the singular values and singular vectors of a matrix. The singular values of $R$ are the square roots of the eigenvalues of either $RR^T$ or $R^TR$, and the left and right singular vectors are the eigenvectors of $RR^T$ and $R^TR$, respectively.

### H.5 RANK AND MATRIX APPROXIMATION

The rank of a matrix $R$ is determined by the number of non-zero singular values in $\Sigma$. By retaining only the largest singular values and their corresponding singular vectors, it is possible to approximate the original matrix $R$ with a lower-rank approximation. This technique is valuable for tasks such as dimensionality reduction and noise reduction, and it is utilized in our approach.

### H.6 PROPERTIES OF SVD

- The singular values in $\Sigma$ are non-negative and arranged in descending order. - The columns of $U$ and $V$ are orthonormal, forming an orthogonal basis for their respective vector spaces. - The SVD decomposition is unique, except for the sign of the singular values and the order of the singular vectors.

SVD is a powerful matrix factorization technique, offering a concise representation of a matrix while preserving essential structural properties. Its applications span various fields, including data analysis, image processing, recommendation systems, and more (Deisenroth et al., 2020).

## I PRINCIPAL COMPONENT ANALYSIS FOR MEAN VARIANCE COMPUTATION

In this section, we outline the procedure for computing the mean variance of high-dimensional data features using Principal Component Analysis (PCA). Consider a dataset $X \in \mathbb{R}^{n \times d}$, where $n$ is the number of data samples and $d$ is the number of features.

### I.1 STANDARDIZATION OF FEATURES

PCA is sensitive to the scale of the input data, so we begin by standardizing the features. This ensures that each feature has zero mean and unit variance.

Let the dataset $X = \{X_1, X_2, \ldots, X_n\}$, where each $X_i \in \mathbb{R}^d$, represent the set of $n$ data samples, each having $d$ features. The standardized data matrix $X_{\text{standardized}}$ is computed as:

$$\mu_j = \frac{1}{n} \sum_{i=1}^{n} X_{ij}, \quad \forall j = 1, 2, \ldots, d, \tag{6}$$

$$X_{\text{centered}} = X - \mu, \tag{7}$$

$$X_{\text{standardized}} = \frac{X_{\text{centered}}}{\sigma}, \quad \sigma_j = \sqrt{\frac{1}{n} \sum_{i=1}^{n} (X_{ij} - \mu_j)^2}. \tag{8}$$

Here, $\mu_j$ represents the mean of the $j$-th feature, and $\sigma_j$ is the standard deviation of the $j$-th feature.

### I.2 COVARIANCE MATRIX COMPUTATION

Once the data is standardized, the covariance matrix $\Sigma \in \mathbb{R}^{d \times d}$ can be computed to measure the pairwise dependencies between features. The covariance matrix is defined as:

$$\Sigma = \frac{1}{n-1} X_{\text{standardized}}^{\top} X_{\text{standardized}}, \tag{9}$$

where $\Sigma_{jk}$ represents the covariance between the $j$-th and $k$-th features.

### I.3 EIGENVALUE DECOMPOSITION OF THE COVARIANCE MATRIX

We perform eigenvalue decomposition on the covariance matrix $\Sigma$, which gives us the principal components and the amount of variance explained by each. The decomposition is expressed as:

$$\Sigma = V \Lambda V^{\top}, \tag{10}$$

where $V \in \mathbb{R}^{d \times d}$ is the matrix of eigenvectors (principal components) and $\Lambda = \text{diag}(\lambda_1, \lambda_2, \ldots, \lambda_d) \in \mathbb{R}^{d \times d}$ is the diagonal matrix of eigenvalues $\lambda_j$, where $\lambda_j$ corresponds to the variance explained by the $j$-th principal component.

### I.4 EXPLAINED VARIANCE

The eigenvalues $\lambda_j$ provide the variance explained by each corresponding principal component. The proportion of variance explained by the $j$-th principal component is computed as:

$$\text{Explained Variance Ratio} = \frac{\lambda_j}{\sum_{k=1}^{d} \lambda_k}. \tag{11}$$

### I.5 MEAN VARIANCE EXPLAINED

The mean variance explained by the principal components can be computed by averaging the explained variance ratio across all components:

$$\text{Mean Variance} = \frac{1}{d} \sum_{j=1}^{d} \frac{\lambda_j}{\sum_{k=1}^{d} \lambda_k}. \tag{12}$$

This metric represents the average amount of variance explained by each principal component in the transformed space.

