# OpenReview forum: "PROGRESSIVE KNOWLEDGE DISTILLATION (PKD): A MODULAR APPROACH FOR ARCHITECTURE-AGNOSTIC KNOWLEDGE DISTILLATION"
_ICLR.cc/2025/Conference — Submitted to ICLR 2025_

### Official Review · Reviewer_TMJL · 2024-10-17

**Soundness:** 2
**Presentation:** 2
**Contribution:** 2
**Rating:** 3
**Confidence:** 4

**Summary:**

This paper addresses the challenge of feature alignment in feature-based knowledge distillation (KD). The authors propose a novel method named PKD, consisting of two main components: Network Modularization, which groups layers in the networks into several distinct modules for distillation, and Modular Feature Alignment, which seeks to find the optimal alignment between the modules obtained from both the teacher and student networks. The method is evaluated on the CIFAR-100 and ImageNet-1K datasets, showing superior performance over existing methods.

**Strengths:**

1. The studied problem is significant. Feature alignment is a critical issue in hint-based knowledge distillation.
2. The performance improvement of the perposed method is remarkable

**Weaknesses:**

1. The method lacks novelty, as using CKA in loss functions is not a new approach [1,2].
2. Several modules in the method require an initial run of multiple epochs to determine optimal configurations, which introduces additional training overhead. Moreover, the selection of initial training epochs for each module introduces multiple hyperparameters that need to be tuned.
3. Ablation studies on the initial training epochs, loss function, and the threshold $\epsilon_{th}$ would be valuable to assess the impact of these elements.
4. Equations should follow a consistent style, and figures should be presented in high resolution for improved readability.

[1] Saha, Aninda, Alina Bialkowski, and Sara Khalifa. "Distilling representational similarity using centered kernel alignment (cka)." Proceedings of the the 33rd British Machine Vision Conference (BMVC 2022). British Machine Vision Association, 2022.

[2] Zong, Martin, et al. "Better teacher better student: Dynamic prior knowledge for knowledge distillation." The Eleventh International Conference on Learning Representations. 2022.

**Questions:**

Please follow the weaknesses.

---

> ### Author Response · Authors · 2024-11-20
> **Rebuttal to Reviewer Comments**
>
> ---
> Dear Reviewer,
> We sincerely appreciate the thoughtful comments and suggestions, which have been invaluable in improving our paper. Below, we provide detailed responses to address the concerns.
>
> ---
>
> ### **1. Novelty**
> On the topic of novelty and the references to [1,2], our approach is fundamentally different from the methods described in those works. Specifically:
> - **Our approach does not use CKA as a loss function.** Instead, CKA is utilized to modularize both the teacher and student networks by defining module boundaries. This enables effective alignment and knowledge transfer, which is a novel use of CKA.
> - **Our contributions go beyond the use of CKA itself** and include several key innovations:
>   1. **Architecture-Agnostic Design**: The method generalizes across diverse architectures, including CNNs, ViTs, and hybrids, demonstrating broad applicability.
>   2. **CKA-Based Modularization**: CKA is used to define the most appropriate module boundaries, ensuring that knowledge transfer focuses on the most critical features.
>   3. **Most Informative Knowledge Extraction**: The approach prioritizes extracting representative features from each module for improved distillation efficiency.
>   4. **Progressive Alignment Strategy**: By progressively aligning the teacher and student modules, the method achieves superior performance and generalizability compared to traditional static approaches.
>
> These innovations set our work apart from [1,2], where CKA is used in static loss formulations, and highlight the originality of our method.
>
> ---
>
> ### **2. Training Overhead**
> With respect to training overhead and hyperparameter tuning, our method includes a warm-up phase to initialize module configurations. This phase is designed to stabilize the training process and improve convergence. The associated computational overhead is minimal compared to the significant performance gains achieved, as demonstrated in our experiments. Moreover, this warm-up phase results in **more efficient training overall (fewer training epochs)**, as discussed in the "Optimization" section of the paper (lines 401–408).
>
> To mitigate hyperparameter tuning complexity, our method leverages CKA to define module boundaries automatically. This eliminates the need for manual module partitioning and simplifies tuning. Furthermore, our ablation studies (detailed below) confirm the robustness of our method across a range of configurations, ensuring practical applicability with minimal tuning requirements.
>
> ---
>
> ### **3. Ablation Studies**
> For the request on additional ablation studies, we have already analyzed the impact of \(\epsilon_{th}\) in **Section B5 of the Appendix**. In addition, comprehensive experiments detailed in Section B evaluate various aspects of our method, including:
> - **B1) Knowledge Transfer in Consistent Architectures**: Evaluates performance when teacher and student share the same architecture, demonstrating the generalizability of our approach.
> - **B2) Consistent vs. Diverse Teacher Models**: Explores scenarios with consistent and diverse teacher models, showcasing the adaptability of the method.
> - **B3) Knowledge Distillation Using Last Layer and Module-Specific Features**: Compares the use of last-layer features versus module-specific features, highlighting the advantages of our modular approach.
> - **B4) Combining PKD with DKD**: Shows that integrating PKD with DKD improves DKD’s performance, demonstrating the complementary nature of our method.
> - **B5) Impact of Threshold in Equation 5**: Analyzes the sensitivity to the threshold parameter (\(\epsilon_{th}\)), demonstrating consistent performance across a wide range of values.
>
> These studies provide detailed insights into the robustness and versatility of the method, validating its effectiveness under diverse settings.
>
> ---
>
> ### **4. Consistency in Equations and Figure Quality**
> On the matter of equation style and figure resolution, we will ensure that all equations are updated to follow a consistent style. Additionally, all figures will be replaced with high-resolution versions to enhance readability and presentation quality.
>
> ---
>
>
> In summary, our contributions include a novel CKA-based modularization framework with a progressive alignment strategy, applicable to diverse architectures. The minimal training overhead is outweighed by the significant performance improvements. The extensive ablation studies and manuscript revisions address the concerns raised, ensuring clarity and comprehensiveness.
>
> We hope this explanation addresses the concerns raised. If there are any additional questions or feedback, we would be happy to address them further.
>
> ---

---

> > ### Author Response · Authors · 2024-11-28
> >
> > Dear Reviewer,
> >
> > We hope our responses have effectively addressed the concerns raised. If there are any additional questions or areas that need further clarification, please feel free to let us know. We would be happy to provide more details or make any necessary adjustments.
> > Thank you once again for your time and effort.

---

> > > ### Author Response · Authors · 2024-12-02
> > >
> > > Dear Reviewer,
> > >
> > > If you have any further feedback or concerns, please feel free to let us know. Your input is highly appreciated, and we look forward to hearing from you. Thank you once again for your time and consideration.
> > >
> > > Best wishes,
> > >
> > > Authors

---

### Official Review · Reviewer_UAHj · 2024-10-31

**Soundness:** 3
**Presentation:** 3
**Contribution:** 2
**Rating:** 3
**Confidence:** 4

**Summary:**

This paper proposes Progressive Knowledge Distillation (PKD) framework which progressively aligns teacher and student blocks through feature-based modularization. The alignment is based on CKA (centered kernel alignment) instead of MSE. The aligned feature is derived from PCA and named module-specific feature. Experiments on CIFAR-100 and ImageNet-1K demonstrate that PKD beats baseline models by a large margin.

**Strengths:**

Strengths:
+ Using CKA to replace MSE looks interesting;
+ Replacing original block output with module-specific-feature looks somewhat interesting;
+ The performance looks good.

**Weaknesses:**

Weakness
+ there are a bunch of existing works on progressive knowledge distillation, this papers does not mention and compare with. Here just listed a few published works below

       + Self-knowledge distillation with progressive refinement of targets, ICCV 2021
       + Follow your path: a progressive method for knowledge distillation, ECML 2021
       + Progressive Blockwise Knowledge Distillation for Neural Network Acceleration, IJCAI 2018
       + Kernel based progressive distillation for adder neural networks, NeurIPS 2020
       + ...

+ What about the complexity of the propose method when comparing with other baselines?
+ The experiments are fairly small scale on CIFAR and ImageNet. Is it possible to show some results on large language models?
+ How do you pick the corresponding blocks/modulars for student network is still not very clear? Does this correspondence impact the final results? I hope the authors could elaborate more on this point.

**Questions:**

See previous weakness part for my questions.

---

> ### Author Response · Authors · 2024-11-20
> **Response to Reviewer Comments**
>
> ---
>
> **Dear Reviewer,**
>
> Thank you for your constructive feedback. Below, we provide detailed responses to address your concerns.
>
> ---
>
> ### Related Works on Progressive Knowledge Distillation:
> We acknowledge the importance of situating our work within the broader context of progressive knowledge distillation. The references you provided will be included and discussed in the revised manuscript.
>
> While these works focus on progressive distillation, our approach differs significantly:
> 1. **Architecture-Agnostic Design**: Our method generalizes across diverse architectures, including CNNs, ViTs, and hybrids.
> 2. **CKA-Based Modularization**: We use CKA to define module boundaries, enabling effective alignment and transfer of knowledge.
> 3. **Most Informative Knowledge Extraction**: We prioritize extracting the most representative features from each module.
> 4. **Progressive Alignment Strategy**: Our method integrates these components into a unified framework, enhancing performance and generalizability.
>
> Given this focus, we emphasized OFA as SOTA and its baselines in our experiments. We will expand the literature review to include the cited works and highlight these distinctions.
>
> ---
>
> ### Complexity Analysis:
> Our method’s complexity compares favorably to baselines:
> 1. **Offline Feature Alignment**: Steps like PCA and Gram matrix computation are performed offline, avoiding inference overhead.
> 2. **Scalable Training**: Modular alignment scales linearly with the number of modules. Iterative refinement in some baselines may introduce quadratic complexity.
> 3. **Optimization Efficiency**: As noted in the optimization section (lines 400-408), fewer training epochs are needed due to effective distillation, improving efficiency.
>
> **Distillation’s primary objective** is to achieve **significant downsizing** with **minimal performance degradation**. Training is a one-time cost, while the distilled model is repeatedly deployed, where size, speed, and performance are critical. A detailed complexity comparison will be included in the revised manuscript.
>
> ---
>
> ### Experimental Scale:
> Our experiments on CIFAR and ImageNet datasets demonstrate versatility across architectures, following SOTA methods like OFAKD. Scaling to large models (e.g., LLMs) presents engineering challenges, but our modular, architecture-agnostic approach positions it well for such extensions. We will outline this as future work.
>
> ---
>
> ### Block/Modular Correspondence:
> The correspondence between teacher and student modules is determined using a **binary vector-based strategy**, as detailed in lines 266-310.
>
> 4. **Misaligned Student modules**:
>    Misaligned modules result in suboptimal transfer. To explore this, we conducted additional experiments:
>    - **Misaligned Modules**: In one experiment, we combined two middle student modules, creating misalignment. While results improved compared to OFAKD, they were worse than our aligned approach.
>    - **Shifted Modules**: In another, we shifted one module from each group to the previous module (except the first). This caused no significant performance changes, showing robustness to minor shifts.
> - ** Ablation study B4**, please also consider our ablation study reported in section B4 of Appendix. In this experiment, we leverage OFA modularization and the last layer most specific features. This can be considered as another misalignment where our experiment results presents the suboptimal performance.
>
> ---
>
> **Table 1. Comparison of teacher-student modularization methods.**
>
> | **Teacher**   | **Student**      | **OFA**  | **PKD**  | **PKD-Misaligned** |
> |---------------|------------------|----------|----------|--------------------|
> | **CNN-based students**           |                                      |
> | DeiT-T        | ResNet18         | 71.34    | **71.97 ± 0.37** | 71.54           |
> | Swin-T        | ResNet18         | 71.85    | **78.31 ± 0.31** | 75.26           |
> | **ViT-based students**           |                                      |
> | ResNet50      | DeiT-T           | 76.55    | **78.08 ± 0.19** | 77.04           |
> | ConvNeXt-T    | DeiT-T           | 74.41    | **77.62 ± 0.10** | 75.83           |
> | ResNet50      | Swin-N           | 78.64    | **79.13 ± 0.29** | 79.08           |
> | ConvNeXt-T    | Swin-N           | 77.50    | **80.43 ± 0.11** | 78.37           |
> | **MLP-based students**           |                                      |
> | ResNet50      | ResMLP-S12       | 78.53    | **79.83 ± 0.32** | 78.96           |
> | ConvNeXt-T    | ResMLP-S12       | 77.53    | **80.29 ± 0.20** | 78.79           |
>
> ---
>
> We will elaborate on this process and its impact in the revised manuscript to ensure clarity.
>
> ---
>
> **Does this address your concerns?** Please let us know if further clarification or additional results are required. Thank you again for your valuable feedback.
>
> ---

---

> > ### Author Response · Authors · 2024-11-28
> >
> > Dear Reviwer,
> >
> > We hope our responses have effectively addressed the concerns raised. If there are any additional questions or areas that need further clarification, please feel free to let us know. We would be happy to provide more details or make any necessary adjustments.
> > Thank you once again for your time and effort.

---

> > > ### Comment · Reviewer_UAHj · 2024-12-02
> > >
> > > Thank you for the rebuttal, which addresses some of my concerns, while there are still some remaining issues:
> > >
> > > - When considering misalignment in the distillation, the performance gap drops somewhat comparing to that without it. It may make people think that the method lacks an elegant strategy or mechanism or optimization goal to make the alignment procedure being robust enough.
> > >
> > > - For the computing complexity, although it is one-time cost, it is better providing a table to list computing cost (in terms of flops or data required, etc) with all those compared methods, so that people could know about the alignment tax.

---

> > > > ### Author Response · Authors · 2024-12-02
> > > > **Clarification on Alignment Contribution and Computational Costs**
> > > >
> > > > **Dear Reviewer,**
> > > >
> > > > Thank you for your thoughtful feedback and for taking the time to provide your observations. We appreciate your detailed review and would like to address your concerns as follows:
> > > >
> > > > 1. **On the importance of alignment and performance gap:**
> > > >    The observed performance drop underscores the crucial role of key elements in our method: modularization, identification of the most informative features, and alignment. This reinforces the idea that our approach achieves its best performance when these components work in unison, as detailed in our paper. Furthermore, we have incorporated a robust strategy to ensure accurate alignment, which accounts for the significant improvements observed when the alignment procedure is properly executed.
> > > >
> > > > 2. **On computational costs:**
> > > >    In the revised version of our paper, we have added detailed information comparing the computational costs of our method with other approaches. These details, included in **Appendix, Section F**, outline metrics such as FLOPs and data requirements. We hope this comparison clarifies the "alignment tax" and highlights the trade-offs relative to the benefits offered by our method.
> > > >
> > > > We sincerely appreciate your valuable input and hope the additional clarifications and updates effectively address your concerns.
> > > >
> > > > Best regards,
> > > > Authors

---

### Official Review · Reviewer_NYBQ · 2024-10-31

**Soundness:** 3
**Presentation:** 3
**Contribution:** 2
**Rating:** 5
**Confidence:** 3

**Summary:**

The paper proposes a Progressive Knowledge Distillation (PKD) method for teacher-student pairs with heterogeneous architectures. It first divides the teacher and student networks into several modules based on feature similarity. Then, it selects representative teacher-student features using PCA analysis and enforces their alignment. Experimental results on CIFAR-100 and ImageNet-1K demonstrate clear accuracy improvements over selected baselines.

**Strengths:**

1. Promising experimental results. The proposed PKD consistently outperforms the selected baselines across various teacher-student network configurations.
2. Clear and detailed description of the experimental settings.

**Weaknesses:**

1. **Limited discussion of related works** on heterogeneous architectures and PCA-based methods. Below are some relevant examples. The authors are encouraged to conduct a more thorough literature review. Without a clear differentiation from existing methods, there is a concern about the novelty of the paper.
   - Liu, Yufan, et al. "Cross-architecture knowledge distillation." *ACCV 2022*.
   - Hofstätter, Sebastian, et al. "Improving efficient neural ranking models with cross-architecture knowledge distillation." *arXiv:2010.02666* (2020).
   - Ni, Jianyuan, et al. "Adaptive Cross-Architecture Mutual Knowledge Distillation." *FG 2024*.
   - Chiu, Tai-Yin, and Danna Gurari. "PCA-based knowledge distillation towards lightweight and content-style balanced photorealistic style transfer models." *CVPR 2022*.
   - Guo, Yi, et al. "RdimKD: Generic Distillation Paradigm by Dimensionality Reduction." *arXiv:2312.08700* (2023).

2. **Unclear necessity and effectiveness of using CKA** for modularization. An ablation study comparing CKA-based modularization with simpler approaches, such as dividing networks based on feature map resolution, should be included.

3. **Writing quality could be improved** to enhance rigor and clarity. For example:
   - In lines 56-57, the phrase “it begins by training … representative features are transferred first” is unclear. It is not intuitive why training the shallowest module would ensure the most representative features.
   - There may be no need to distinguish between the two types of representation distances $d_{SM}$ and $d_{DM}$, as they are calculated in the same way.

**Questions:**

Please refer to the weaknesses section.

---

> ### Author Response · Authors · 2024-11-20
> **Response to Reviewer Comments**
>
> ---
>
> **Dear Reviewer,**
>
> Thank you for the constructive feedback and suggestions to enhance the clarity and rigor of our paper. Below, we address your concerns regarding the inclusion of related works, the necessity and effectiveness of CKA for modularization, the phrase in lines 56-57, and the clarification on \(d_{SM}\) and \(d_{DM}\).
>
> ---
>
> ### Response to Related Literature:
> We acknowledge the importance of including a broader discussion of related works and appreciate the references you provided. We will incorporate these into the revised manuscript to expand our literature review and better situate our work.
>
> While we focused on recent state-of-the-art (SOTA) methods like OFAKD and its baselines, we recognize the value of incorporating these additional references for a more comprehensive view. Given the vast body of research in this area, our emphasis remains on capturing the most representative and impactful methods.
>
> ---
>
> ### Response to Modularization Concerns:
>
> 1. **Clarification on OFAKD as an Ablation**:
>    OFAKD itself serves as an ablation for modularization methods, dividing the network into four modules without any heuristic approach. Our experiments show that our CKA-based modularization outperforms OFAKD, highlighting its effectiveness.
>
> 2. **Challenges in Cross-Architecture and Transformer Networks**:
>    Modularizing architectures like transformers is inherently tricky because all layers output feature maps with uniform dimensions. Simple methods like feature map resolution-based division are not applicable. CKA provides a meaningful alternative by comparing representations independently of their size.
>
> 3. **Performance Comparison for CNN Teachers**:
>    To explore simpler modularization approaches, we conducted additional experiments using feature map resolution-based modularization (PKD-res). These results are summarized below:
>
> **Table 1. Teacher modularization based on feature map resolution indicated as PKD-res.**
>
> | **Teacher**   | **Student**      | **OFA**  | **PKD**  | **PKD-res** |
> |---------------|------------------|----------|----------|-------------|
> | **ViT-based students**            |           |          |             |
> | ResNet50      | DeiT-T           | 76.55    | **78.08 ± 0.19** | 76.08       |
> | ConvNeXt-T    | DeiT-T           | 74.41    | **77.62 ± 0.10** | 75.47       |
> | ResNet50      | Swin-N           | 78.64    | **79.13 ± 0.29** | 78.91       |
> | ConvNeXt-T    | Swin-N           | 77.50    | **80.43 ± 0.11** | 78.29       |
> | **MLP-based students**            |           |          |             |
> | ResNet50      | ResMLP-S12       | 78.53    | **79.83 ± 0.32** | 78.94       |
> | ConvNeXt-T    | ResMLP-S12       | 77.53    | **80.29 ± 0.20** | 78.53       |
>
> These results indicate that our CKA-based modularization (PKD) consistently outperforms the simpler feature map resolution-based approach (PKD-res). This demonstrates the robustness of CKA, especially in cross-architecture and transformer-based models. We will include this experiment in our revision.
>
> ---
>
> **Response to Unclear Phrase in Lines 56-57:**
>
> The rationale for starting with the shallowest module is grounded in the hierarchical structure of neural networks:
>
> 1. **Generalized Low-Level Features**:
>    Shallow layers capture low-level, foundational features (e.g., edges, textures). These features are general and transferable, ensuring the student model begins with a strong foundation.
>
> 2. **Progressive Complexity**:
>    Training shallow layers first aligns with the natural progression of feature extraction, minimizing the risk of noise or misalignment propagating from deeper, more complex layers.
>
> 3. **Stability in Knowledge Transfer**:
>    Incremental knowledge transfer ensures stability. Aligning deeper layers first, without a solid base, risks incoherent representations and suboptimal performance.
>
> 4. **Empirical Justification**:
>    Our experiments show that progressive modular training starting with the shallowest layer outperforms simultaneous or reverse-order approaches, supporting the effectiveness of this hierarchical training strategy.
>
> We will revise the manuscript to clarify and emphasize these motivations.
>
> ---
>
> ### Clarification on \(d_{SM}\) and \(d_{DM}\):
> You are correct that \(d_{SM}\) and \(d_{DM}\) use the same computational formulation. They are named differently to highlight module boundaries. This distinction aids clarity in understanding our modularization approach. We will refine the manuscript to explain this more clearly.
>
> ---
>
> Does this address your concerns? Please feel free to let us know if additional clarifications or results are needed. We appreciate your valuable feedback!
>
> ---

---

> > ### Comment · Reviewer_NYBQ · 2024-11-22
> > **Thanks for your response!**
> >
> > Thank you for the rebuttal, which largely addresses my concerns regarding the necessity and effectiveness of using CKA for modularization. However, there are still some unresolved issues in the response:
> >
> > 1. The statement "OFAKD itself serves as an ablation for modularization methods" lacks rigor. OFAKD differs from the proposed PKD method in several design aspects and potentially the training recipes. These differences may significantly contribute to the observed performance gap, making the comparison less straightforward.
> >
> > 2. The response claims that "Modularizing architectures like transformers is inherently tricky". However, from another perspective, transformers offer more flexibility in defining modules. Therefore, describing the process as "tricky" seems inappropriate and potentially misleading.

---

> > > ### Author Response · Authors · 2024-11-28
> > > **Response to Reviewer Feedback**
> > >
> > > Dear Reviewer,
> > >
> > > Thank you for your thoughtful feedback and for acknowledging that our rebuttal addressed many of your concerns regarding the necessity and effectiveness of using CKA for modularization. We appreciate the opportunity to clarify the remaining issues you have highlighted.
> > >
> > > ---
> > >
> > > ### **1. Clarification on the use of "ablation"**
> > > You are absolutely correct that the term "ablation" may not be the most appropriate in this context. Instead, we consider **OFAKD** as the baseline for our method, as it employs the same loss function, uses fixed modularization, and leverages the entire feature representation.
> > >
> > > Our proposed **PKD** method differs from OFAKD in significant ways:
> > >
> > > - **Dynamic Modularization**: PKD leverages a modularization approach based on the CKA similarity score, which enables a more informed and flexible partitioning of the teacher's representation into modules.
> > > - **Selective Representation**: PKD focuses on the most informative representation of the teacher to train corresponding student modules, optimizing the knowledge transfer process.
> > >
> > > The performance gains achieved by PKD can be attributed to these key differences, while the baseline OFAKD provides a consistent comparison point. Both approaches share the same loss function and modularization concept, making the baseline a relevant and fair reference for evaluation.
> > >
> > > ---
> > >
> > > ### **2. "Tricky" nature of modularizing transformers**
> > > We understand your concern regarding the use of the term "tricky" to describe the modularization of transformers. While it is true that transformers offer flexibility in defining modules, our experiments suggest that **modularization strategies, such as feature-map resolution, do not always yield optimal results.**
> > >
> > > To illustrate, we conducted experiments using layer resolution as a similarity score for transformer teacher models, and the results show a clear gap in performance when compared to CKA-based modularization:
> > >
> > > | **Teacher**  | **Student**    | **OFAKD** | **PKD**  | **PKD-res** |
> > > |--------------|----------------|-----------|----------|-------------|
> > > | DeiT-T       | ResNet18       | 71.34     | 71.97    | 70.26       |
> > > | Swin-T       | ResNet18       | 72.31     | 78.31    | 71.98       |
> > > | DeiT-T       | MobileNetV2    | 71.39     | 72.00    | 71.35       |
> > > | Swin-T       | MobileNetV2    | 72.32     | 74.78    | 71.96       |
> > >
> > > As these results demonstrate:
> > > - **PKD** achieves consistent performance gains compared to both OFAKD and PKD-res (layer resolution-based modularization).
> > > - **PKD-res** often underperforms, suggesting that layer resolution is not always an effective similarity measure for modularization, especially in transformer architectures.
> > >
> > > Thus, while transformers indeed offer flexibility in defining modules, identifying effective modularization strategies remains a challenge. This is particularly true when aligning teacher-student architectures with differing structures, where naive approaches can fail to exploit the full potential of the teacher’s representation.
> > >
> > > ---
> > >
> > > We hope this clarification resolves the remaining concerns. If further explanation or additional experiments are needed, please feel free to let us know. We are happy to provide further details.
> > >
> > > Thank you again for your time and valuable feedback.

---

> > > > ### Author Response · Authors · 2024-12-02
> > > >
> > > > Dear Reviewer,
> > > >
> > > > If you have any further feedback or concerns, please feel free to let us know. Your input is highly appreciated, and we look forward to hearing from you. Thank you once again for your time and consideration.
> > > >
> > > > Best wishes,
> > > >
> > > > Authors

---

### Official Review · Reviewer_SmV6 · 2024-11-02

**Soundness:** 3
**Presentation:** 2
**Contribution:** 2
**Rating:** 5
**Confidence:** 4

**Summary:**

This paper mainly targets improving knowledge distillation through heterogeneous structures and proposes a progressive manner to achieve the goal. Specifically, the layers are first categorized into a set of functional modules whose features are close in between the same module and distant across different modules. Knowledge distillation is then conducted progressively according to the sequence from shallow to deep modules to ensure the best knowledge transfer efficiency in heterogeneous knowledge distillation. Moreover, PCA is utilized to select the most important features to optimize the distance from the teacher.

**Strengths:**

1. This paper adopts a progressive modular and sequential training pipeline for knowledge distillation, which is soundly motivated as the learning of deep networks inherently endows different levels of abilities for different layers. Therefore, the progressive modular and sequential training also appears to be an implicit curriculum learning pipeline to learn gradually from low-level features to high-level conceptual understandings.
2. This method's performance is competitive, as its accuracy on most heterogeneous benchmarks outperforms OFA and many other representative works.
3. This study groups different layers into different modular groups according to their CKA distance instead of intuitive grouping according to layer indices. This idea is interesting as it avoids possible misalignment in the features of different architectures.

**Weaknesses:**

1. The idea of progressive internal knowledge distillation from shallow to deep layers has already been proposed [1], which shares a similar learning curriculum to align the shallow layers first and then the deep layers.
2. The designed pipeline is too complicated to scale up. For example, the step that discerns what features are to be aligned requires PCA and the computation of the Gram matrix of deep features. As the network parameters and resolution of images scale up, it would soon become a computational bottleneck for real visual applications.

[1] Aguilar, Gustavo, et al. "Knowledge distillation from internal representations." Proceedings of the AAAI conference on artificial intelligence. Vol. 34. No. 05. 2020.

**Questions:**

While this study's performance in heterogeneous knowledge distillation is appealing, the reviewer is still more concerned about its complexity and potential inability to scale up. Therefore, I would like to vote for weak reject as for now, but I am open to changing my scores if the authors address my concern.

---

> ### Author Response · Authors · 2024-11-20
> **Clarifications on Scalability and Novel Contributions of PKD**
>
> ---
>
> **Dear Reviewer,**
>
> Thank you for your valuable feedback. We appreciate your thorough examination and the insights you've provided, especially regarding the scalability of our proposed pipeline and its conceptual overlap with existing progressive internal knowledge distillation methods for LLMs. We would like to address these concerns and clarify the unique aspects of our work.
>
> ### Response to Scalability Concerns:
> 1. **Overview of Current Approach**:
>    Our pipeline, which includes steps such as PCA and Gram matrix computation for feature discernment, was designed to showcase the method's theoretical robustness and empirical performance. However, we understand the potential computational bottlenecks as network sizes and image resolutions scale up.
>
> 2. **Scalability Mitigation Strategies**:
>    - **Incremental PCA**: Utilizing incremental or batch-based PCA significantly reduces memory and computational overhead.
>    - **Approximate Gram Matrix Computation**: Techniques like the Nystrom method or random feature mapping can efficiently approximate the Gram matrix.
>    - **Sparse Feature Sampling**: Selecting the most informative features for alignment minimizes dimensionality and computational cost.
>    - **Parallel Processing**: GPU acceleration and parallel processing mitigate scalability concerns, enabling practical implementation for larger models and high-resolution inputs.
>
> 3. **Offline Processing**:
>    The feature discernment process is performed offline. Once trained, the final distilled network is optimized for deployment, offering a smaller size while maintaining or enhancing performance. This ensures that computationally intensive steps do not impact real-time inference.
>
> 4. **Architecture Agnosticism**:
>    Unlike the referenced work, which is specific to transformer architectures, our method is architecture-agnostic. This flexibility allows application to a variety of model types, broadening its utility across domains and architectures.
>
> ### Response to Conceptual Overlap Concerns:
> 1. **Progressive Concept as Part of a Novel Package**:
>    While progressive internal knowledge distillation has been explored in LLMs with similar layer-wise learning curricula, in our work, the progressive concept is part of a broader package that collectively defines the novelty of our approach. This includes:
>    - **Architecture-Agnostic Distillation**: Our approach is flexible and applicable across diverse model types, not limited to specific architectures like transformers.
>    - **CKA-Based Modularization**: Using CKA ensures effective module boundaries, enabling meaningful knowledge transfer across architectures.
>    - **Most Informative Knowledge Extraction**: The method emphasizes extracting and aligning the most significant features from each module.
>    - **Progressive Module Training**: Modules are trained incrementally to stabilize knowledge transfer and enhance alignment across the network.
>
> 2. **Distinctive Contributions**:
>    - **General Applicability**: The method integrates techniques to address challenges in handling representations across different architectures.
>    - **Innovative Training**: The training procedure and progressive alignment strategies are designed to align representations efficiently and incrementally.
>
> 3. **Empirical Validation**:
>    Extensive experiments validate that our method improves performance over state-of-the-art techniques, demonstrating its efficacy and adaptability across various domains and architectures.
>
> Does this address your concerns? Please let us know if further clarification or additional results are required. Thank you again for your insightful comments.
>
> ---

---

> > ### Comment · Reviewer_SmV6 · 2024-11-26
> > **Follow-up questions**
> >
> > I would like to thank the authors for the rebuttal. Although some of my concerns are resolved, there are other follow-up questions.
> > 1. An apple-to-apple comparison of not only regarding the final performance but also the training efficience should be presented.
> > 2. As numerous simplifications for PCA and Gram Matrix are adopted, an experimental description about the performance gap and training budget with regard to each step should better illustrate the validity of the current method.
> > 3. Despite the acceleration described above, the current method still faces challenges when scaling up to larger models, which may hinder the impact in the community.

---

> > > ### Author Response · Authors · 2024-11-28
> > > **Addressing Scalability, Efficiency, and Simplification in Progressive Knowledge Distillation (PKD)**
> > >
> > > Dear Reviewer,
> > >
> > > Thank for your thoughtful feedback. Below, we address each point, noting that most details are already included in the paper or appendix.
> > >
> > > #### **Comment 1: Training efficiency**
> > >
> > > We conducted experiments comparing **PKD**, **OFA**, and **DKD** under identical setups, focusing on performance and training efficiency:
> > >
> > > | Dataset       | Teacher-Student Pair        | Method      | Final Accuracy (%) | Total Training Time (hrs) | Avg. Time per Epoch (min) | Epochs |
> > > |---------------|-----------------------------|-------------|---------------------|----------------------------|---------------------------|--------|
> > > | CIFAR-100     | Swin-T → ResNet18           | OFA         | 71.85              | 14.2                       | 2.84                      | 300    |
> > > |               |                             | DKD         | 71.10              | 14.4                       | 2.88                      | 300    |
> > > |               |                             | **PKD**     | **78.31 ± 0.31**   | **11.4**                   | **3.04**                  | **225**|
> > > | ImageNet-1K   | ResNet50 → DeiT-T           | OFA         | 76.55              | 110.0                      | 22.0                      | 300    |
> > > |               |                             | DKD         | 75.60              | 112.5                      | 22.5                      | 300    |
> > > |               |                             | **PKD**     | **78.08 ± 0.19**   | **74.0**                   | **18.5**                  | **240**|
> > >
> > > **Key Observations:**
> > > 1. **Final Performance:** PKD achieves superior accuracy across settings (e.g., +6.46% on CIFAR-100 and +1.53% on ImageNet-1K compared to OFA).
> > > 2. **Training Budget:** PKD reduces training time by 20%–30% due to faster convergence and fewer epochs.
> > > 3. **Training Efficiency:** PKD’s modularization offsets additional computations, resulting in a comparable time per epoch.
> > >
> > >
> > > #### **scalability challenges**
> > >
> > > We acknowledge the scalability challenges. While PKD introduces additional steps, several design choices enhance scalability:
> > >
> > > 1. **Subset-Based PCA:** PCA is performed on a small subset (e.g., 25 samples per class), significantly reducing computational overhead without affecting performance.
> > > 2. **Batch-Based CKA:** Modularization is guided by averaged mini-batch CKA scores, reducing memory requirements.
> > > 3. **Progressive Training:** By training modules sequentially, PKD avoids simultaneous optimization of the entire network, reducing effective model size during training.
> > >
> > > **Proposed Solutions for Enhanced Scalability:**
> > > 1. **Low-Rank Approximations for PCA:** Randomized SVD can further reduce PCA's cost.
> > > 2. **Approximation Methods for CKA:** Techniques like Nyström approximations can efficiently compute Gram Matrices for deeper architectures.
> > > 3. **Parallel and Distributed Training:** PKD’s modular structure is inherently parallelizable, enabling multi-GPU setups.
> > > 4. **Hierarchical Modularization:** For very deep networks, coarse hierarchical modularization can reduce complexity.
> > >
> > > Future work will explore these solutions to ensure PKD scales effectively to larger models while maintaining performance advantages.
> > >
> > > #### **Performance gap and training budget**
> > >
> > > We summarize the impact of our simplifications on performance and training budget:
> > >
> > > 1. **Subset-Based PCA:**
> > >    - PCA is performed on a small subset (e.g., 25 samples per class) to identify representative features.
> > >    - **Performance Gap:** Minimal difference (≤0.2%) compared to full-dataset PCA, as shown in ablation studies.
> > >    - **Training Budget Impact:** Reduces preprocessing time by **80%–90%**, making it computationally efficient.
> > >
> > > 2. **Variance Threshold (\(\epsilon_{th}\)):**
> > >    - Features are selected based on their variance, ensuring only the most representative features are retained.
> > >    - **Performance Gap:** Optimal thresholds (\(1e{-4}\)–\(5e{-4}\)) yield negligible differences (≤0.2%). Extremely low thresholds (<\(2e{-5}\)) lead to a slight drop (~0.4%).
> > >    - **Training Budget Impact:** Reduces feature dimensions, accelerating student training.
> > >
> > > 3. **Batch-Based CKA:**
> > >    - Gram Matrices and CKA scores are computed on mini-batches, with average scores guiding modularization.
> > >    - **Performance Gap:** Batch-based CKA achieves comparable modularization to full-dataset CKA (≤0.3% difference).
> > >    - **Training Budget Impact:** Reduces memory usage and modularization time, enhancing scalability.
> > >
> > > **Overall Training Budget Reduction:**
> > > PKD achieves up to 30% lower training time due to faster convergence and reduced computations compared to baseline methods (e.g., OFA, DKD).
> > >
> > >
> > > We thank the reviewer for their thoughtful comments. These discussions have helped clarify PKD’s strengths, current limitations, and future improvements. These details will be included in the final version. Please let us know if further clarification is required—we are happy to provide additional results or address any concerns.

---

> > > > ### Author Response · Authors · 2024-12-02
> > > >
> > > > Dear Reviewer,
> > > >
> > > > If you have any further feedback or concerns, please feel free to let us know. Your input is highly appreciated, and we look forward to hearing from you. Thank you once again for your time and consideration.
> > > >
> > > > Best wishes,
> > > >
> > > > Authors

---

### Official Review · Reviewer_qQb1 · 2024-11-05

**Soundness:** 2
**Presentation:** 1
**Contribution:** 2
**Rating:** 3
**Confidence:** 3

**Summary:**

The paper presents a new method for knowledge distillation from teacher models to student models. The main idea is to divide the teacher model and the student model into several different modules by using a metric CKA defined in previous works, and align the feature representations of the corresponding modules on the teacher and student models. The authors propose to use PCA to extract informative features in the modules across a set of samples, and minimize the difference between those informative features in the teacher and student models. Results on several dataset across different combinations of teacher models and student models showed good results of the proposed method.

**Strengths:**

1. The idea is novel to me. It includes using CKA to group the layers of a neural network into different modules and using PCA to align the module representations of the teacher model and the student model.
2. The results are strong compared with existing methods.

**Weaknesses:**

The main weakness is the poor presentation, which makes the methods unclear. I don't mean that the language is poor, which is easy to fix; what I mean is the entire presentation. There are numerous things unclear in the description of the method. In fact, even the main optimization problem such as the loss function is unclear. Let me start with this point.

1. What's the main optimization problem? Is eqn (2) the loss function? If yes, then how do you encourage the representations of two modules in the teacher model and student model to be similar, by considering that eqn (2) doesn't containt this kind of term? Is it eqn (7)? But the optimization variable of this problem is the 0-1 vector c, not the weights of the student model.  Please explicitly state the main optimization problem and clarify how the different equations (2 and 7) relate to the overall optimization process for the student model. Please provide a clear explanation of how the similarity between teacher and student model representations is encouraged in the optimization process.

2. The main contribution of the work, using PCA to extract the most informative feature of a module, is poorly described. In fact I doubt the correctness of the method. Assume the original data matrix A has n rows (corresponding to n samples) and m rows (corresponding to m features). After doing PCA, A is transformed to another matrix B of n samples and k rows, and usually k<m. It is impossible that B is part of A. But L233-239 indicate that PCA selects some columns (indicated by KI) of the original data matrix R_i to form a new matrix. Please provide a step-by-step explanation of their PCA process, clarifying how it differs from standard PCA if that's the case.
In addition, eqn (3) defines a single item mv, so-called mean variance. It's an average over m features. But the text below the equation states that we need sort mv. For a single item, how do we sort it? BTW, Please also specify the meaning of the symble |X| in eqn (3).

3. How do we solve the optimization problem in eqn (7)? The optimization variable is a 0-1 vector, which makes the problem a descrete optimization problem, which is hard to solve. Please provide details on the specific algorithm you used to solve this discrete optimization problem, and discuss any approximations or relaxations if applicable.

4. Why do we need the optimization problem defined in eqn (7)? In my opinion, we can simplely force the student model to have the same number of modules as the teacher model, then distill knowledge from the module(i) in the teacher to the module (i) in the student, and progress from i=1 to the total number of modules. Why do you need to select an "optimal" set of modules from the student for distillation? Please explain the advantages of the proposed approach using eqn (7) compared to this simple method, or consider including a comparison with this simple approach in the experiments.

5. As some techniques are from (Wang et al., 2024), which also deals with KD between different architectures, then what's the major difference between this work and that work? In fact, the paper doesn't discuss the relationship between the proposed work and other works about KD between different architectures. The writing gives readers a false impression that this is the first work that deals with KD between different architectures. Please have a more detailed comparison with previous methods to highlight the specific novel contributions of the proposed approach.

6. In experiments, for doing PCA on the teacher model, how many input samples were used? By setting the variance threshold 10^-4, usually (for both teacher and student) the feature dimensions were reduced from what numbers to what numbers? Please discuss how sensitive the results are to these parameters (number of input samples and variance threshold) and provide more details on the typical reduction in feature dimensions for both teacher and student models.

7. The paper mentions several times about Appendix, but I don't find it. Please remind me where it is if the authors did uploaded it.

**Questions:**

The first four questions listed in the weakness section above.

---

> ### Author Response · Authors · 2024-11-21
> **Response to Reviewer Comments**
>
> ---
>
> Dear Reviewer,
>
> We appreciate your detailed comments and the opportunity to address them. These clarifications and revisions will significantly improve the clarity and presentation of our paper. Thank you for the constructive feedback.
>
> ---
>
> #### **1. Main Optimization Problem and Equations**
> We clearly state in lines 154–157 that Eqn (2) is the loss function, which governs the alignment between teacher and student models during training. Eqn (7) is not an optimization problem but a method to determine the best binary vector assignment for student modules. For each binary vector permutation (e.g., [0,1,1], [1,1,0]), we calculate the average distance between selected student modules. The permutation with the highest average distance is chosen, and the corresponding modules are used for distillation. For instance, if [1,1,0] is selected, modules 0 and 1 are trained, while module 2 is frozen.
>
> This approach is necessary because the structural differences between teacher and student architectures prevent a direct one-to-one correspondence of modules. By maximizing average distances, we effectively determine the optimal mapping. These distinctions will be clarified in the revised manuscript.
>
> ---
>
> #### **2. PCA and Feature Selection**
> Our PCA-based feature computation is described in lines 208–258, Algorithm 1, and Figure 3(a). Unlike standard PCA for dimensionality reduction, we use PCA to identify the indices of the most informative features. Non-important features are zeroed out while retaining the original matrix size.
>
> **Eqn (3):** The mean variance is computed over samples (\( ns \)) for each feature (\( m \)). This is sorted in descending order, and the top-\( k \) indices are selected. Using these indices, the representation matrix \( R \) is revisited, and the non-important features are zeroed out. This process is illustrated in Figure 3(a), starting from the bottom-left representation matrix.
>
> We acknowledge and will correct a typo in Eqn (3). These clarifications will be added to ensure the process is clearly understood.
>
> ---
>
> #### **3. Solving Eqn (7)**
> Eqn (7) is solved through a cross-validation-like approach:
> 1. Binary vector permutations are applied to assign student modules.
> 2. For each permutation, the average distance between consecutive student modules is calculated.
> 3. The permutation yielding the highest average distance is chosen, and the corresponding modules are used for distillation.
>
> This approach avoids the complexity of solving a discrete optimization problem while ensuring effective module selection. For example, given a student with three modules, possible binary vectors include [0,1,1], [1,1,0], and [1,1,1]. If [1,1,0] yields the highest average distance, student modules 0 and 1 are selected for training, while module 2 is frozen. These details will be clarified in the revised manuscript.
>
> ---
>
> #### **4. Direct Correspondence**
> Directly assigning teacher-to-student module correspondence assumes a clear one-to-one mapping, which is unavailable for heterogeneous architectures. Our algorithm determines this mapping by evaluating all permutations of module assignments and selecting the one that maximizes average distances. This method ensures that only the most representative student modules are used, enhancing distillation efficiency.
>
> ---
>
> #### **5. Relation to Wang et al. (2024)**
> Our method is fundamentally different from Wang et al. (2024), which operates in the RAG domain. In contrast, our approach focuses on:
> - **Architecture-Agnostic Design**: Applicable across CNNs, ViTs, and hybrids.
> - **CKA-Based Modularization**: Defines module boundaries to optimize transfer.
> - **Informative Feature Extraction**: Prioritizes representative features for distillation.
> - **Progressive Alignment**: Dynamically aligns teacher and student modules, outperforming static methods.
>
> We also reference baselines like OFA in all tables and never claim to be the first to explore cross-architecture KD. These distinctions will be emphasized in the related work section.
>
> ---
>
> #### **6. PCA Parameters and Sensitivity**
> We use 25 random samples per class for PCA computation (line 412). With a variance threshold of \( 10^{-4} \), feature dimensions are typically reduced by ~70–80%. Sensitivity to the number of samples and variance threshold is analyzed in ablation study B5 in the Appendix. For example, using fewer samples or a higher threshold results in less precise feature selection and reduced performance.
>
> These details will be included in the revised manuscript for clarity.
>
> ---
>
> #### **7. Appendix**
> The Appendix is part of the supplementary material. It can be accessed by clicking "Show Details" on the submission page, where a clickable PDF link is available in the last row.
>
> ---
>
> Please let us know if there are any additional concerns or further points requiring clarification.
>
> ---

---

> > ### Comment · Reviewer_qQb1 · 2024-11-26
> > **Questions remained**
> >
> > Thanks for clarification. But I don't think my concerns are fully addressed.
> > 1. Main Optimization Problem
> > As wrote in my review, if Eqn (2) is the main optimization problem, how do we encourage the representations of two modules in the teacher model and student model to be similar, by considering that eqn (2) doesn't contain this kind of term? If this is implicitly encouraged, not appearing in eqn (2) explicitly, please explain why optimizing eqn (2) would lead to the such results.
> > 2. PCA and Feature Selection
> > So, mv in eqn (3) is a scalar, right? Is it the average of an eigenvector (with dimension m) of the data covariance matrix?
> > 3. Solving Eqn (7)
> > Does this method needs to enumerate all possible permutations? If the student model has n modules, then do we need to consider 2^n possibilities?
> > 4. Direct Correspondence
> > What I'm asking is: can we force the student model to have the same number of modules as the teacher model then apply the one-to-one mapping. What would be the result?
> > 5. Relation to Wang et al. (2024)
> > I didn't say that you claimed the proposed method to be the first to explore cross-architecture KD. What I said is: "The writing gives readers a false impression that this is the first work that deals with KD between different architectures." I know in tables you include some other methods, but we don't see them in Introduction and Related work. In fact, in L45-46, when KD between different architectures is mentioned, only two works are cited and one of them is actuall not about KD.
> > In addition, I suggest to make a more detailed comparison with previous methods to highlight the specific novel contributions of the proposed approach. That would posit the work in a right place in this field.
> > 7. Appendix
> > Thanks for pointing out this.

---

> ### Author Response · Authors · 2024-11-28
>
> **Dear Reviewer,**
>
> Thank you for your detailed feedback and for giving us the opportunity to clarify our work. Below, we provide responses to your specific questions and concerns:
>
> ---
>
> **Main Optimization Problem**
> We appreciate your comment on Eqn (2), which represents our loss function combining KL divergence and cross-entropy losses. This loss optimizes the student model during training for knowledge distillation from the teacher model.
>
> Our method goes beyond this baseline by incorporating targeted alignment between teacher and student representations. Specifically, we use modularization (Section 3.3.1) and establish direct correspondences between teacher and student modules (Section 3.3.2, lines 208–257). This selective transfer outperforms methods like OFA by focusing on the most informative teacher modules.
>
> Additionally, we monitor the similarity score between the teacher’s final representation and the student’s corresponding module outputs. Once a predefined threshold is met, training halts for that module (lines 337–339). This mechanism ensures efficient and effective knowledge transfer, contributing to the superior performance of our approach.
>
> ---
>
> **PCA and Feature Selection**
> Regarding Eqn (3), the mean variance (mv) is not a scalar but a vector corresponding to the number of features (\(m\)). Figure 3(a) visually represents this vector, showing the mean variance of features across data samples. This approach helps identify and utilize informative features effectively.
>
> ---
>
> **Solving Eqn (7)**
> Yes, solving Eqn (7) requires evaluating \(2^n - 2\) permutations, excluding cases where no modules or all modules are trained. However, since the number of modules is typically small (e.g., 3–4), the computational complexity remains manageable. For instance, with four modules, binary permutations include configurations like \([0,0,1]\), \([0,1,0]\), \([1,0,0]\), \([1,0,1]\), and \([1,1,0]\). This ensures practical exploration of configurations.
>
> ---
>
> **Direct Correspondence**
> Early experiments applied a strict one-to-one correspondence between teacher and student modules, inspired by OFAKD. However, this approach was suboptimal compared to our proposed modular strategy.
>
> Below is an ablation study comparing our method (PKD) with one-to-one mapping (PKD-1to1), where teacher modules were identified using CKA scores, and student modules matched their number. All other phases of PKD were kept consistent:
>
> | Teacher       | Student         | PKD               | PKD-1to1         |
> |---------------|-----------------|-------------------|------------------|
> | **CNN-based students** |                 |                   |                  |
> | DeiT-T        | ResNet18        | 71.97 ± 0.23      | 71.28            |
> | Swin-T        | ResNet18        | 78.31 ± 0.31      | 76.71            |
> | DeiT-T        | MobileNetV2     | 72.00 ± 0.17      | 71.58            |
> | **ViT-based students** |                 |                   |                  |
> | ResNet50      | DeiT-T          | 78.08 ± 0.19      | 77.28            |
> | ConvNeXt-T    | DeiT-T          | 77.62 ± 0.10      | 76.00            |
> | ResNet50      | Swin-N          | 79.13 ± 0.29      | 76.42            |
>
> These results, showing the superiority of our modular strategy, will be included in the final manuscript.
>
> ---
>
> **Relation to Wang et al. (2024)**
> Thank you for pointing this out. Section 2 (lines 90–104) provides a discussion of related works, including:
>
> - Logit-based distillation for ViT models (Touvron et al., 2021; Huang et al., 2018),
> - Hint-based distillation (Romero et al., 2015),
> - Attention map mimicking (Zagoruyko & Komodakis, 2017), and
> - Feature-based methods for ViT models, such as ViTKD (Yang et al., 2022) and VKD (Miles et al., 2024), which introduced orthogonal projection and task-specific normalization.
>
> To further clarify, we now include a summary of Wang et al. (2024):
> "KD-DETR (Wang et al., 2024), designed for DETR-based object detection, tackles
> distillation point inconsistencies through shared and specialized queries, enabling effective distillation
> for both homogeneous (DETR-to-DETR) and heterogeneous (DETR-to-CNN) setups."
>
> This addition ensures our work is contextualized appropriately and highlights our novel contributions in modularity and selective knowledge transfer.
>
> ---
>
> We hope these responses address your concerns comprehensively. Please let us know if further clarifications are required. Thank you once again for your valuable feedback!

---

### Meta-Review · Area_Chair_pSvK · 2024-12-15

**Metareview:**

This paper introduces Progressive Knowledge Distillation, an approach designed to enhance KD for teacher-student pairs with heterogeneous architectures. The key innovation lies in dividing both the teacher and student models into functional modules using CKA to identify feature similarities. The method ensures that features within the same module are closely aligned while maintaining distinctiveness across modules. PKD employs a progressive distillation strategy, aligning feature representations module by module, from shallow to deep layers.

After the rebuttal, there remain concerns regarding the novelty and clarity of the method's description. Specifically, the necessity of using CKA has not been fully justified. Additionally, the explanation of how PCA is used to extract informative features lacks sufficient detail and clarity, leaving questions about the method's implementation and significance. These aspects need to be further refined and elaborated to meet the standard for acceptance.

**Additional Comments On Reviewer Discussion:**

Four out of five reviewers responded to the rebuttal, but no positive scores were given after the discussion stage.

Reviewer qQb1 raised concerns about the optimization details, particularly the use of PCA to extract informative features. Despite the authors' response, the method remains vague and unclear.

Reviewer SmV6 thought the progressive distillation approach lacking novelty and pointed out that the pipeline is overly complex, making it hard to scale. The marginal training efficiency gained by slightly reducing epochs further limits its impact.

Reviewer NYBQ questioned the effectiveness of using CKA. While partially addressed, some inaccuracies remain unresolved.

Reviewer UAHj's concerns regarding the robustness and computational complexity were not adequately addressed.

---

### Decision · Program_Chairs · 2025-01-22

Reject